

# The probability distribution of daily streamflow in the conterminous United States

Annalise G. Blum [1], Richard M. Vogel[1], Stacey A. Archfield [2]

[1]Civil and Environmental Engineering, Tufts University, Medford, MA, 02155, USA

[2]U.S. Geological Survey, Reston, VA, 20192, USA

*Correspondence to*: Richard M. Vogel (Richard.Vogel@tufts.edu)

**Abstract.** One of the most commonly used tools in hydrology, empirical flow duration curves (FDCs) characterize the frequency with which streamflows are equaled or exceeded. Finding a suitable probability distribution to approximate a FDC enables regionalization and prediction

of FDCs in basins that lack streamflow measurements. FDCs constructed from daily streamflow observations can be computed as the period-of-record FDC (POR-FDC) to represent long-term streamflow conditions or as the median annual FDC (MA-FDC) to represent streamflows in a typical year. The goal of this study is to identify suitable probability distributions for both POR-FDCs and MA-FDCs of daily streamflow for unregulated and

perennial streams. Comparisons of modeled and empirical FDCs at over 400 unregulated stream gages across the conterminous United States reveal that both the four-parameter kappa (KAP) and three-parameter generalized Pareto (GPA3) distributions can provide reasonable approximations to MA-FDCs; however, even four and five-parameter distributions are unable to capture the complexity of POR-FDC behavior, often with flows ranging over five or more

orders of magnitude. Regional regression models developed for the mid-Atlantic and Missouri regions as case studies present a simple and practical method to predict MA-FDCs at ungaged sites, which can be accurately predicted more consistently compared to POR-FDCs.



# 1 Introduction

One of the most commonly used tools in hydrology, flow duration curves (FDCs) illustrate the frequency with which streamflows are equaled or exceeded. Applications of FDCs include allocation of water, wastewater management, hydro-power assessments, sediment

transport, protection of ecosystem health (Castellarin et al., 2013; Vogel and Fennessey, 1995) and the generation of time series of daily streamflows at ungaged sites (Archfield and Vogel, 2010). Traditionally, FDCs have been constructed for the full period of observed record (POR-FDCs) by ranking flows from all recorded years and plotting them against an estimated empirical exceedance probability. POR-FDCs are particularly useful for representing the long-

term or steady-state hydrologic regime at a site, and for estimation of time-series of daily streamflows at ungaged sites (Archfield and Vogel, 2010; Fennessey, 1994; Hughes and Smakhtin, 1996).

In contrast to a POR-FDC, mean and median annual FDCs represent the frequency of flows in a typical year and are less dependent upon the specific period of record (LeBoutillier

and Waylen, 1993; Vogel and Fennessey, 1994). Mean or median annual FDCs are derived from a series of annual FDCs (AFDCs), a ranking of all flows within a single year. The full set of AFDCs can also be used for the construction of confidence intervals (Castellarin et al., 2007; Serinaldi, 2011; Vogel and Fennessey, 1994). For representing the frequency and magnitude of streamflows in a typical year, median annual FDCs (MA-FDCs) are generally preferred to

mean annual FDCs because they are less influenced by abnormally dry or wet years (Vogel and Fennessey, 1994).

MA-FDCs are increasingly used as an alternative to POR-FDCs when flows in a typical year are of primary interest. For example, MA-FDCs have recently been used to predict hydropower production (Mohor et al., 2015; Müller et al., 2014), evaluate regional similarity

between streams under different flow conditions (Patil and Stieglitz, 2011), and characterize



baseflow variability (Hamel et al., 2015). MA-FDCs are also used to compare streamflow regimes in different catchments (Hrachowitz et al., 2009) or assess before and after watershed land-use changes (Kinoshita and Hogue, 2014). Commonly used in ecological assessment of rivers, MA-FDCs can help quantify fish passage delays (Lang et al., 2004). Median seasonal

FDCs have been used to evaluate impacts on ecological flow regimes and to define the concepts of ecodeficit, ecosurplus, and ecochange (Gao et al., 2009; Lin et al., 2014; Vogel et al., 2007).

### 1.1 *Prediction of FDCs in ungaged basins*

Broad regions of the world have little or even no records of streamflows, making prediction of flows in ungaged basins a major challenge (Sivapalan et al., 2003). Many studies

have developed methods to predict POR-FDCs at ungaged sites. For an extensive review of these methods, refer to Chapter 7 in "Runoff prediction in ungaged basins" (Castellarin et al., 2013) and to Archfield and Vogel (2010). Process-based models are an increasingly popular method of deriving and predicting FDCs at ungaged basins because they offer the ability to relate watershed physical characteristics to streamflow regimes. While promising for locations

without streamflow data, process-based FDC models require numerous assumptions regarding runoff and climate mechanisms (Botter et al., 2008; Yokoo and Sivapalan, 2011).

The most commonly used methods to estimate POR-FDCs in ungaged basins are statistically based, such as regression and index flow methods. Castellarin et al. (2004) introduced a promising index-flow model to predict FDCs at ungaged sites which links AFDCs

to the POR-FDC. Many statistical methods, including index flow approaches, depend upon the identification of a suitable probability distribution to fit to the POR-FDC. This remains a considerable challenge because daily streamflows often range over many orders of magnitude thus exhibiting extraordinary positive skewness. Nevertheless, if an appropriate probability distribution can be identified, regional regression models of the distribution parameters can be

used to estimate daily flows at ungaged sites. In identifying a suitable probability distribution,



the need for sufficient parameters to describe the complex distribution of daily streamflows must be balanced with the challenge that additional parameters hinder parameter identifiability, estimation and interpretation (Castellarin et al., 2007). Given the difficulty of selecting a single distribution to approximate the probability distribution of daily streamflows, some studies have

focused on a portion of the FDC, such as flows below the median (Fennessey and Vogel, 1990) or above the mean (Segura et al., 2013) or have considered separate models for wet and dry seasons (Bowers et al., 2012).

### 1.2 *Probability distributions to approximate FDCs*

Vogel and Fennessey (1993) and Fennessey (1994) concluded that a 3-parameter

generalized Pareto (GPA3) distribution provides a suitable approximation to the probability distribution of POR-FDCs in Massachusetts and the northeastern U.S. respectively, based upon L-moment diagrams. When additional goodness-of-fit metrics were considered, the GPA3 was rejected as a suitable probability distribution for POR-FDCs in Italy (Castellarin et al., 2004). Multiple authors have noted that a complex distribution with at least four parameters is needed

to approximate the probability distribution of daily streamflows (Archfield, 2009; Castellarin et al., 2004; LeBoutillier and Waylen, 1993). Castellarin et al. (2007) selected the 4-parameter Kappa distribution (KAP) to approximate the distribution of daily streamflows for a region in Italy. For the northeastern U.S., Archfield (2009) also found KAP, and, to a lesser degree, GPA3, to provide a good approximation for POR-FDCs; nevertheless a number of challenges

were identified in fitting the KAP and GPA3 to the tails of the POR-FDCs. For the lowest flows, Archfield (2009) found that the application of both the GPA3 and KAP probability distributions generated negative streamflows - a physical implausibility - at some sites and at other sites led to consistent over-estimation of minimum flows or under-estimation of maximum flows due to probability distribution theoretical lower and upper bounds,

respectively. Even more complex probability distributions than the GPA3 and KAP considered



by Archfield (2009) were not promising either; for example, the five-parameter Wakeby distribution could not be fit at over half the study sites considered.

A few studies have focused on fitting a probability distribution to FDCs for a typical year rather than the whole period of record. LeBoutillier and Wayland (1993) found a 5-parameter mixed lognormal distribution to be superior to 2- and 3-parameter lognormal, Gamma and generalized extreme value distributions for fitting probability distributions to mean annual FDCs of four rivers in Canada. Fennessey (1994) is the only study to our knowledge to fit a probability distribution to median annual FDCs. For the mid-Atlantic U.S., Fennessey investigated a number of 2 and 3-parameter distributions and identified the GPA3 as a suitable distribution for both POR- and MA-FDCs. In addition, he developed regional regression models to relate GPA3 model parameters to measurable basin and climate characteristics and then used those models to estimate future impacts of climate change on FDCs.

In this study, comprehensive goodness-of-fit evaluations are used to assess the ability of several probability distributions to approximate the distribution of both POR-FDCs and MA-FDCs of daily streamflows. The study region includes over 400 gaged, unregulated, perennial watersheds across the conterminous US. To our knowledge, no study has previously attempted to evaluate the GOF of alternative probability distributions to daily FDCs for as large a region as the conterminous U.S. The paper is organized as follows. We begin with a description of the methods to construct POR- and MA-FDCs, the candidate probability distributions, and an introduction to the study region. Next we provide results: L-moment ratio diagrams, quantitative GOF of probability distributions and example flow duration curves illustrating the range of fits. Based on these results, we develop regional models for estimation of FDCs at ungaged sites for two case study regions. Finally, we summarize our findings and provide directions for future research.





## 2 Methods and study region

### 2.2 *POR- FDC and MA-FDC estimation*

A FDC is defined as the complement of the cumulative distribution function:

$$1 - F_Q(q), \text{ where } F_Q(q) = P\{Q \le q\} \tag{1}$$

where q represents observed streamflow and $F_Q(q)$ is the empirical cumulative distribution function of observed streamflow. The first step in constructing a POR-FDC is to rank the flows, $q_i$, in ascending order as in $q_{(1)}...q_{(365n)}$ where *n* is the number of years of record. For leap years, flows from February 29 are removed to maintain consistent sample sizes across years. Next, these flows are plotted against a plotting position, which is an estimate of the exceedance

probability associated with each ordered observation. The Weibull plotting position is used as it provides an unbiased estimate of exceedance probability, regardless of the underlying probability distribution of the ranked observations (Vogel and Fennessey, 1994):

$$P\{Q > q\} = 1 - \frac{i}{365n+1} \tag{2}$$

where *i* represents the rank. Vogel and Fennessey (1994) review several alternative and more

complex nonparametric quantile estimators that can be used to construct FDCs; this simple estimator is selected here given the large sample sizes (365 for each MA-FDC and at least 365×40 for each POR-FDC). The procedure for constructing an MA-FDC is similar, but rather than ranking all of the 365×*n* flows, flows are ranked within each calendar year resulting in *n* lists of rankings 1-365 for each of the *n* years. Then the median flow at each ranking is selected.

Figure 1 illustrates of the differences between the POR-FDC, AFDCs and the MA-FDC at an example USGS streamflow gage. The majority of the POR- and MA-FDC curves are very similar, but differ at the lowest and highest durations because the most extreme streamflows on record are always included within POR-FDCs yet those same extreme flows do not impact




the construction of MA-FDCs because the median estimator is insensitive to outliers. See Vogel and Fennessey (1994) for a more detailed discussion of this issue.

### 2.1 *Candidate probability distributions and selection*

We build off of previous work suggesting the KAP and GPA3 distribution as candidate

5 probability distributions (Fennessey,1994; Castellarin et al., 2007; Archfield, 2009). The GPA3, generalized extreme value and generalized logistic distributions are all special cases of the KAP distribution and the exponential distribution is a special case of the GPA3 distribution (Hosking, 1994, 1997). The quantile function, *x(F)*, the inverse of the cumulative distribution function, *F(x)*, for a GPA3 distribution is given by Hosking and Wallis (1997):

$$x(\mathrm{F}) = \begin{cases} \xi - \alpha \log(1 - \mathrm{F}(x)), & k = 0 \\ \xi + \frac{\alpha}{k}\left[1 - (1 - \mathrm{F}(x))^k\right], & k \neq 0 \end{cases} \qquad (3)$$

where $\xi$ is the location parameter, $\alpha$ is a scale parameter, and *k* is a shape parameter. When k=0, GPA3 simplifies to the exponential distribution. The KAP distribution includes the same three parameters as GPA3 plus an additional shape parameter, h. The quantile function of the KAP distribution is given by Hosking (1994):

$$x(F) = \xi + \frac{\alpha}{k}\left[1 - \left(\frac{1 - F(x)^h}{h}\right)^k\right] \qquad (4)$$

where log represents the natural log.

As an initial assessment, L-moment diagrams were used to identify candidate probability distributions for both POR- and MA-FDCs. L-moments are linear combinations of probability weighted moments and are known to be more robust to outliers and less biased than

20 ordinary product moment ratios (Hosking, 1990). L-moment diagrams provide a visual method of comparing the GOF of various probability distributions to observed data. Vogel and Fennessey (1993) demonstrate that L-moment diagrams are often superior to ordinary moment diagrams, particularly for very long records of highly skewed samples of daily streamflows, as



is the focus of this study. Even when parent distributions are complex, L-moment diagrams are useful in identifying simpler distributions that fit the observed data sufficiently well (Stedinger et al., 1993). For a description of the theory of L-moments, see Hosking (1990). To visually identify candidate distributions, sample estimates of L-kurtosis and L-skew ratios are compared

with their theoretical relationships using L-moment diagrams.

### 2.3 *Study region*

Because human activities can have substantial impacts on FDCs (Castellarin et al., 2013), only gages "unregulated" by humans were included (Falcone, 2011)  in our analysis. Unregulated gages in the conterminous US with at least 40 years of daily mean streamflow

records since 1950 from the USGS Hydro-Climatic Data Network (HCDN-2009) dataset (USGS, 2009) were selected for inclusion to minimize impacts due to differences in sampling variability between sites (Vogel et al., 1998). Some previous studies have focused on fitting a probability distribution to daily streamflows at small and/or intermittent streams (Mendicino and Senatore, 2013; Pumo et al., 2014). Here, 170 sites having an average daily flow value of

zero (flows below 0.01 feet$^3$/second) were dropped from analysis because they require more complex methods to fully capture the distribution of streamflow.

For the resulting 420 gages, mean daily streamflows were obtained from the USGS National Water Information System (U.S. Geological Survey, 2001). Figure 2 shows the location of the gages within the US. Record lengths of study sites range from 40-61 years

between 1950-2010. Drainage areas of the basins associated with each gage vary from 1.5 to over 14,000 km$^2$. Basin characteristics were obtained from the GAGES-II (Geospatial Attributes of Gages for Evaluating Streamflow) dataset (Falcone, 2011), which were available for 398 of the 420 sites. These basin characteristics were used to develop regional regression equations for two case study regions in the US: the Missouri region (HUC #10) and the mid-

Atlantic region (HUC #02).





## 3 Results

### 3.1 *Initial identification of candidate distributions using L-moment diagrams*

Figure 3 shows curves representing theoretical relationships between L-kurtosis and L-skew ratios of several three-parameter distributions compared with sample estimates of L-moment ratios computed from empirical POR- and MA-FDCs at study sites. On these L-moment diagrams, three-parameter distributions are shown as curves, the four-parameter Kappa distribution is represented by the area between the curve of the generalized logistic (GLO) and the theoretical lower bound for L-kurtosis given L-skewness for all distributions (ALL.LB), and the lower bound of the five-parameter Wakeby distribution (WAK.LB) is given by a curve. The L-moment diagrams for the POR-FDC (Fig. 3, left) and MA-FDC (Fig. 3, right) appear to be similar, which is expected given that the points represent L-moment ratios for the same 420 US sites. However, as we show below, there is greater sampling variability associated with the estimated L-moment ratios corresponding to the MA-FDCs due to their much smaller sample sizes (365) compared with the sample sizes associated with the POR-FDCs which equal $365 \times n$.  The bulk of the points in both plots in Fig. 3 appear to be concentrated around the dashed GPA3 line, suggesting that GPA3 is the most promising 3-parameter distribution, among the probability distributions considered, as expected from the literature. Given a sufficiently long record length, we would expect the points in Fig. 3 to fall on the GPA3 theoretical curve if distribution of daily streamflows arose from the GPA3 distribution. The fact that the points fill an area bounded roughly by the GLO curve and the ALL.LB indicates that the four parameter KAP distribution may also provide a suitable fit to both POR-FDCs and MA-FDCs. We additionally investigate the 5-parameter Wakeby distribution (WAK).

The scatter of points around the GPA3 theoretical curve in Fig. 3 could be due to sampling variability resulting from limited sample sizes. To assess whether limited record



length explains the scatter, synthetic daily streamflows sequences were generated from the GPA3, KAP and WAK distributions where each sequence had the same record length as the site upon which its L-moments are based. These synthetic data were simulated using the following steps: (1) compute the sample L-moments for each site from the observed data; (2)

estimate the distribution parameters associated with the respective probability distribution from the sample L-moments for GPA3, KAP, or WAK; (3) simulate data of the same record length; and (4) compute L-moment ratios from the simulated sample and plot the ratios on an L-moment diagram. Of the 420 sites, KAP parameters could not be estimated due to sample L-moment ratios that were inconsistent with the KAP distribution at 36 sites (9%) for POR-FDCs

and 20 sites (5%) for MA-FDCs. These sites with L-moments inconsistent with KAP tended to have smaller drainage areas and flows, however there did not appear to be a clear pattern. Only 158 of the 420 sites (38%) had daily flows which could be fit with WAK using L-moments, a result found in another study that had attempted to fit WAK to daily streamflows (Archfield, 2009). GPA3 parameters were valid at all sites for both FDCs, an advantage of the GPA3

distribution for this application.

The impact of sampling variability on POR-FDCs is shown in Fig. 4, which illustrates L-moment ratios from data simulated by GPA3 (left), KAP (center) and WAK (right) probability distributions. Nearly all of the GPA3-simulated POR-FDC L-moment ratios (grey crosses) fall on the GPA3 line. If the observed flows arose from a GPA3 process with the large

sample sizes considered here, we would also expect minimal scatter resulting from sampling variability around the theoretical GPA3 curve in Fig. 3. Thus, the scatter in L-moment ratios about the GPA3 line in Fig. 3 does not appear to be due to sampling but rather reflects the complexity of the true distribution from which the daily streamflows arise. Because the L-moment ratios corresponding to the observations do exhibit significant scatter around the

GPA3 curve (Fig. 3), it seems unlikely that those observations arise from a GPA3 process. Compared to GPA3, simulated L-moment ratios from KAP (Fig. 4, center) appear more





consistent with L-moment ratios estimated from empirical POR-FDCs (Fig. 3). At the 158 sites for which WAK parameters could be fit, L-moment ratios simulated from WAK (Fig. 4, right) appear less consistent with empirical L-moment ratios (Fig. 3) compared to ratios simulated from KAP. We conclude from these initial evaluations that the KAP probability distribution

appears to provide the best fit among the probability distributions considered.

Simulation of synthetic samples that mimic empirical MA-FDCs poses a considerable challenge. One could generate each AFDC as a subsample of length 365 from an assumed POR-FDC based on the theory outlined by Castellarin et al. (Castellarin et al., 2007) and Serinaldi (2011) and then compute the MA-FDC of those resulting synthetic AFDCs.

However, arbitrarily dividing a simulated POR-FDC into samples of 365 would not be appropriate because it would ignore the important serial stochastic structure of daily flows, including such issues as autocorrelation and seasonality. To correctly simulate AFDCs from a POR-FDC which, in turn, could be used to estimate an MA-FDC, a more complete understanding of the stochastic structure of such daily flows is needed. To employ the index-

flow model of FDCs proposed by Castellarin et al. (2004) and later extended by Serinaldi (2011) requires assumptions concerning the probability distribution of both the POR-FDC as well as the series of annual streamflows (AF) at each site needed to implement such an index-flow FDC analysis. We show in Appendix A that selection of a suitable probability distribution to approximate both the standardized POR-FDC and the AFs may be even more challenging

than selection of a suitable probability distribution to approximate the POR-FDC. Given the theoretical advantages associated with the index flow method of FDCs, we recommend that future attention be given to these approaches introduced by Castellarin et al. (2004) and Serinaldi (2011). However, in this study, we seek a single suitable probability distribution to approximate the POR-FDC and the MA-FDC for practical applications, thus we do not give

further consideration to the index-flow FDC method.



### 3.2 *Overall goodness of fit evaluations of various probability distributions to daily streamflow series*

Although L-moment diagrams are useful for providing a rapid and approximate overview of the goodness-of-fit (GOF) of promising probability distributions for observed processes,

they are only one tool to evaluate the distributional properties of data. Perhaps the two most commonly-used tools to evaluate the suitability of a model to reproduce observations in hydrology are the standardized mean square error commonly referred to as Nash-Sutcliffe Efficiency (NSE) and percent bias. The most common estimator of NSE at each site is:

$$NSE = 1 - \frac{\sum_{x=1}^{X}(Q_x - Q_x^{pred})^2}{\sum_{x=1}^{X}(Q_x - \overline{Q_x})^2} \tag{5}$$

where $Q_x$ represents observed flow at quantile x $Q_x^{pred}$ predicted flow at quantile x, $\overline{Q_x} = \sum_{x=1}^{X} \frac{Q_x}{X}$ the mean value of the observed flows, and $X$ the total number of daily flows (and therefore number of quantiles). NSE values range from -∞ to a maximum of 1, which would indicate that the estimated values matched observed exactly. Because both NSE and percent bias will be heavily influenced by the highest flows, values are given for the entire FDC as well

as broken down into values above and below the median streamflow. This is a particular issue when assessing the GOF of daily streamflows because they span 5 or more orders of magnitude and, therefore, exhibit such enormous values of skewness as documented by Vogel and Fennessey (1993).

Figure 5 gives boxplots summarizing the NSE for POR-FDCs and MA-FDCs fit with

GPA3 (left plot) and those fit with KAP (right plot). Recall that the method of L-moments (Hosking and Wallis, 1997) was used to estimate parameters of the KAP and GPA3 probability distributions corresponding to the POR- and MA-FDCs. The range of NSE coefficients from the whole POR-FDC is given in white, followed by flows above the median in light grey and then flows below the median flow in dark grey. To enable comparisons, the plots all range from

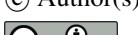



0-1 and some negative values of NSE are not shown (but the number omitted is listed in the figure caption). First, comparing the right plot to the left plot, we conclude that the four-parameter KAP distribution generally performs better than the 3-parameter GPA3 distribution. Next, comparing POR-FDC to MA-FDC NSE coefficients, NSE is much higher for MA-FDC

5    both overall and for flows above the median. The median NSE for flows below the median is similar for both POR-FDC to MA-FDC, however MA-FDC has a slightly lower 25[th] percentile.

As a measure of standardized mean square error, the NSE is made up of both the effects of bias and variance. To separate these two effects, we also examine bias separately. We compute percent bias to account for differences in the magnitudes of the flows across sites:

$$\% \, Bias = 100 * \frac{\frac{1}{X}\sum_{x=1}^{X}(Q_x^{pred} - Q_x)}{\frac{1}{X}\sum_{x=1}^{X} Q_x} \tag{6}$$

In Fig. 6, boxplots illustrate that %bias is very close to zero for both POR-FDC and MA-FDC across the entire FDC as well as above and below the median. We find some slightly larger %bias for flows below the median but the 25[th] and 75[th] percentiles of %bias remains within +/- 10%.

15        As found in other studies, fitted KAP and GPA3 probability distributions generate negative flows and/or exhibit theoretical upper or lower bounds inconsistent with observed flows at some sites (Archfield, 2009; Castellarin et al., 2007). Table B1 (Appendix B) provides counts of these inconsistencies for both MA-FDCs and POR-FDCs and illustrates that these inconsistencies are problematic for both FDCs as well as both GPA3 and KAP probability

20    distributions. In general, the fitted KAP probability distribution had more problems with lower bounds while the GPA probability distribution was less able to accurately reflect the highest flows. Parameter estimates derived to reproduce the expected value of the minimum observed streamflow can be used to ensure a non-negative lower bound of flows for GPA3 (Fennessey, 1994), but are difficult to derive for the KAP probability distribution (Archfield, 2009). Both

25    Castellarin et al. (2007) and Archfield (2009) tried to enforce a lower bound of zero for the





KAP probability distribution, however, Archfield found these methods did not improve the overall fit of the resulting probability distributions. More research is needed on how to obtain parameter estimates for the KAP and GPA3 probability distributions that generate lower and upper bounds consistent with the properties of daily streamflow observations (i.e. streamflow

cannot be negative; the upper bound is infinite).

### 3.3 *Graphical Evaluation of flow duration curves*

Part of the reason why FDCs are so widely used in practice is that they provide a graphical illustration of the complete relationship between the magnitude and frequency of streamflow as was illustrated in Fig. 1. Such FDC plots also enable a very effective visual

comparison of the relative GOF of the GPA3 and KAP distributions for POR- and MA-FDCs as is shown in Fig. 7 and Fig. 8. Visual comparisons provide an opportunity to assess which portions of the FDC the fitted probability distributions are best able to approximate streamflows and also provide a visual assessment of NSE and %bias values. To illustrate overall GOF, Fig. 7 and Fig. 8 focus on three sites representing the lowest (worst fit), median (typical fit), and

highest (best fit) based on values of NSE for the GPA3 probability distribution across all sites in the study.

Starting on the right, both KAP and GPA3 provide an excellent fit to the "best" POR-FDC and MA-FDC plots, as illustrated in Fig. 7 and Fig. 8, respectively. The site with the median NSE for both POR- and MA-FDCs is the same and this site indicates a comparable fit

across the two FDCs. Despite this comparable fit, the NSE coefficients are quite different: 0.89 for POR-FDC GPA3 versus the much higher 0.96 for MA-FDC GPA3. This discrepancy reflects a challenge in the use of the metric and indicates why visual inspection of FDC plots is particularly important for understanding overall GOF. Looking to the plots of sites with the lowest NSE, the POR-FDC is not well fit by the GPA3 and even the KAP led to both over-

estimation of the upper tail and under-estimation of the lower tail. The site with the lowest NSE





for the MA-FDC was one in which KAP parameters could not be estimated because they were inconsistent with the L-moment ratios at that site. The GPA3 distribution, while underestimating the highest flows, nevertheless provides a relatively good fit to the rest of the MA-FDC even for this worst case. We conclude from these evaluations that both the GPA3

and KAP probability distributions provide good models for representing MA-FDCs across the entire U.S., whereas, neither model is able to adequately reproduce observed POR-FDCs for many of the sites considered. This conclusion could only have been reached by a careful assessment of both quantitative and graphical GOF evaluations; using NSE and L-moment diagrams alone would have masked these observations.

### *4. Regional case studies*

Now that we have documented the improved GOF of theoretical probability distributions to MA-FDCs compared with POR-FDCs, it is of interest to explore whether this result can lead to improvements in our ability to estimate FDCs at ungaged sites. Castellarin et

al. (2013) compare a number of methods of estimating FDCs at ungaged sites and note that statistical methods are most common. Regression models for a region with gaged streams can be used to develop models to predict parameters of a probability distribution based on basin characteristics. If the basin characteristics of an ungaged site are known, the regional regression models can be used to estimate the parameters of the probability distribution and simulate a

FDC. Here we illustrate how our findings can be applied to the prediction of FDCs at ungaged sites using the mid-Atlantic and the Missouri Hydrologic Units (HUCs# 02 and 10 respectively; Fig. 2) as case studies. These regions were selected for having a sufficient number of unregulated sites to enable estimation of regional relationships between probability distribution model parameters and watershed characteristics. In addition, both regions exhibit a significant

hydrologic heterogeneity, creating a challenge for estimation of FDCs at ungaged sites. The GPA3 distribution was selected for regionalization because it is a parsimonious model with





three parameters that each has a clear physical interpretation: $\xi$ represents the theoretical lower bound, $\alpha$ the scale, and $\kappa$ the shape.

Leave-one-out cross validation, which effectively treats each stream gage as though it were ungaged, is used to evaluate the regionalization method. Within each of the two regions,

each site was omitted from the data set iteratively and regional regression equations were developed and then applied to the removed stream gage. Power-law regression models for each of the three GPA3 model parameters for POR- and MA-FDCs were developed, resulting in six regression models per region (three models for the POR-FDC and three models for the MA-FDC). These models are based on those used by Fennessey (1994) and include comparable

basin characteristics now available in the GAGESII dataset (Falcone, 2011). The following regional regression models for the three parameters ($\xi$, $\alpha$, and $\kappa$) were fit using ordinary least squares by taking natural log of each equation:

$$(\widehat{\xi + 1}) = \beta_0 DA^{\beta_1} P^{\beta_2} BFI^{\beta_3} AW^{\beta_4} \tag{7}$$

$$\hat{\alpha} = \beta_5 DA^{\beta_6} P^{\beta_7} BFI^{\beta_8} PET^{\beta_9} TOP^{\beta_{10}} \tag{8}$$

$$(\widehat{\kappa + 1}) = \beta_{11} DA^{\beta_{12}} P^{\beta_{13}} WD^{\beta_{14}} \tag{9}$$

where $\hat{\xi}$, $\hat{\alpha}$, and $\hat{\kappa}$ are the predicted values for the three parameters of the GPA3 probability distribution and the βs are the estimated coefficients of each model. The explanatory variables are as follows: DA is drainage area, P is the long-term annual average precipitation for the basin, BFI is the long-term average base flow index, AW is average available water capacity

for soil layer, PET is mean annual potential evapotranspiration estimated using the Hamon equation (Hamon, 1961), TOP is the topographic wetness index and WD is wet days, or the average annual number of days of measurable precipitation (Falcone, 2011). Appendix C provides more detailed descriptions of these variables and estimated coefficients for the regression models for both POR- and MA-FDCs for HUCs 02 and 10. Adjusted $R^2$ values for

the 12 models range from 19% to 98% (Appendix C, Table C2). Regression assumptions of



normality and homoscedasticity were assessed through diagnostic checks of regression residuals. All model residuals appeared to be approximately homoscedastic based upon plots of residuals. Based on the Shapiro-Wilks test, the null hypothesis that the residuals are normally distributed could not be rejected at the 5% level for almost all models (Appendix C, Table C3).

Detailed regression results are not given here as these case studies are only intended to illustrate how our results can be applied but are not the primary focus of this study.

Figure 9 contrasts the GOF of the regional regression approach with results from a corresponding at-site analysis based on the observed data at each site, which was obtained using the results from Sect. 3.1. Here the regional regression is used to fit a GPA3 model to the

observations at each site, and the NSE of all flows (including flows both above and below the median) is used to compare the GOF of flows estimated by fitting a GPA3 to the at-site observations using L-moments. Boxplots of NSE values for at-site results are compared to regional regression models for POR- and MA-FDCs in Fig. 9. Both at site and regional models are able to predict flows within HUC 02 (left) much more consistently compared to HUC 10

(right). This is consistent with other studies which have found GOF higher in the mid-Atlantic region compared to the Missouri region (Newman et al., 2015; Martinez and Gupta, 2010). Overall, NSE values are generally higher for MA-FDCs compared to POR-FDCs in both regions. We conclude from this analysis that regionalization of MA-FDCs should in general lead to more accurate predictions of FDCs than regionalization of POR-FDCs.

**5 Conclusions**

Combining quantitative goodness-of-fit comparisons in addition to graphical assessment based on L-moment diagrams and visual assessment of graphical FDC plots, we conducted an extensive search for a single probability distribution function that could fit all daily streamflows at over 400 perennial, unregulated stream gage sites in the conterminous

United States. We find that both the four-parameter Kappa (KAP) and the three-parameter





generalized Pareto (GPA3) distributions provide very good approximations to median annual flow duration curves (MA-FDCs), however neither distribution provides nearly as suitable a fit to the much more complex and complete distribution of daily streamflows, the period of record flow duration curve (POR-FDC). It seems unlikely that a single theoretical 3-, 4, or 5-

parameter distribution will be able to capture the extraordinary complexity and variability associated with the full range of daily flows exhibited in POR-FDCs. Some caveats with the use of POR-FDCs include that these FDCs can only be considered to yield steady state exceedance probabilities given a sufficiently long period of record; if the period of record is either much shorter than the planning period or includes abnormally dry or wet periods, the

interpretation of POR-FDCs as a steady-state representation of streamflow may be misleading. In general, the interpretation of a POR-FDC is limited to the exact period of record used in its construction. Thus, for certain water resource applications such as prediction of typical hydropower production in a year or managing flows for ecological services, prediction of a MA-FDC might be more useful than POR-FDC. Few previous studies have sought to evaluate

theoretical probability distributions for modelling MA-FDCs which represent streamflow magnitude and frequency during a typical year. However, the many uses of MA-FDCs suggest that our findings could have broad applications. We caution users of MA-FDCs to be aware that they can only provide a window into the behaviour of streamflow in a typical year, thus they should not be used when severe floods and droughts are of interest.

Others have noted the need for models with at least four-parameters to describe the complex distribution of daily streamflows. We find that even a four parameter probability distribution is insufficient to suitably describe POR-FDCs at unregulated perennial streams in the US. Both at-site and regional parameter estimation methods will be subject to even greater challenges in estimation and interpretation of parameters as one considers more complex

probability distributions. We have identified challenges in accurately reproducing the behaviour of minimum observed streamflow due to problems relating to parameter estimation





and the challenge of enforcing the conditions that observed streamflows must be both non-negative and always exceed theoretical distributional lower bounds.

Finally, we performed a small case study to evaluate our ability to develop a regional model of POR- and MA-FDCs for two regions in the US using a 3-parameter probability

distribution. Such models are quite useful for estimation of FDCs as well as time series of streamflows at ungaged sites. Based on leave-one-out cross validation experiments, we find that regional hydrologic models of the parameters of a GPA3 distribution can predict MA-FDCs at ungaged sites quite accurately. However, the prediction of POR-FDCs was less consistent. For applications of FDCs for which interpretation of the magnitude-frequency

relationship in a typical year suffices, prediction of a MA-FDC may provide a more accurate estimate of daily streamflows than attempts to describe the steady-state magnitude-frequency relationship using the POR-FDC.

Daily streamflow varies over four or five orders of magnitude and is subject to seasonality and serial correlation; when viewed though this lens, the finding of any candidate

distribution - such as those explored here - that provides some explanatory power is remarkable. The case study provides evidence that there is potential for use of a single pdf of daily streamflow for some applications. Continued work to assess the feasible upper and lower bounds could bring us closer to understanding the behaviour of FDCs.

**Appendix A. L-moment ratio diagrams for index flow method of constructing regional FDCs**

As this study is closely related to the index-flow method of developing regional models of POR-FDCs and AFDCS (Castellarin et al., 2004) and to the theoretical approach introduced by Serinaldi (2011) to construct confidence intervals for AFDCs, we explore here the problem

of fitting a probability distribution to index flow FDCs. The index-flow method introduced by Castellarin et al. (2004) requires splitting the daily streamflows into two components: (1) the




series of annual flows (AF) and (2) the daily flows divided by the annual flow in each corresponding year which we term the index flows. Figure A1 illustrates L-moment diagrams for these two components. L-Cv vs L-skew diagrams are useful for identifying 2-parameter distributions; a plot comparing these L-moment ratios for the series of AFs is given in the left

panel of Fig. A1. The left plot suggests that the probability distribution of AFs in the US can be approximated by the 2-parameter Gamma distribution, a result which is consistent with other recent national (Vogel and Wilson, 1996) and global studies (McMahon et al., 2007). The right plot in Fig. A1 shows L-kurtosis vs L-skew ratios for the index flows which are simply the daily flows standardized by each year's annual flow (POR-FDC/AF, represented by empty

circles) compared to the POR-FDC (solid grey points). This plot indicates that the distribution of the standardized daily flows is of comparable complexity to that of the unstandardized POR-FDC. We conclude that the theoretical framework introduced by Castellarin et al. (2004) and Serinaldi (2011) while extremely promising, will pose tremendous challenges in terms of finding a suitable probability distribution to model the distribution of index flows in

combination with the AFs.

**Appendix B. Probability distribution function parameter estimation challenges**

Table B1. For fitted generalized Pareto (GPA3) and Kappa (KAP) distributions, counts and percentage of sites with negative predictions or theoretical bounds inconsistent with observed period of record flow duration curves (POR-
FDC) and median annual flow duration curves (MA-FDC) compared to total sites with feasible parameters for each distribution

| | GPA3 | | KAP | |
|---|---|---|---|---|
| *Sites with:* | *POR-FDC* | *MA-FDC* | *POR-FDC* | *MA-FDC* |
| Negative flows predicted | 102/420 (24%) | 88/420 (21%) | 47/384 (12%) | 13/400 (3%) |
| Observed flows>upper bound | 4/10 (40%) | 4/15 (27%) | 7/24 (29%) | 6/75 (8%) |
| Observed flows<lower bound | 265/420 (63%) | 147/420 (35%) | 287/383 (75%) | 243/400 (61%) |
| Theoretical lower bound <0 | 102/420 (24%) | 92/420 (22%) | 58/384 (15%) | 26/400 (7%) |





## Appendix C. Regional regression variables and summary

**Table C1. Description of explanatory variables for regional regression equations. These are all taken from GAGESII database, except for the drainage area of the basin (Falcone, 2011)**

| Variable | Description |
|---|---|
| DA_km2 | Drainage area in (km2) |
| PPTAVG_BASIN | Mean annual precipitation (cm) for the watershed, from 800m PRISM data. 30 years period of record 1971-2000. |
| BFI_AVE | Base Flow Index (BFI), The BFI is a ratio of base flow to total streamflow, expressed as a percentage and ranging from 0 to 100. Base flow is the sustained, slowly varying component of streamflow, usually attributed to ground-water discharge to a stream. |
| PET | Mean-annual potential evapotranspiration (mm/year), estimated using the Hamon (1961) equation |
| TOPWET | Topographic wetness index, log(a/S); where "log" is the natural log, "a" is the upslope area per unit contour length and "S" is the slope at that point |
| WD_SITE | Site average of annual number of days (days) of measurable precipitation, derived from 30 years of record (1961-1990), 2km PRISM. |
| AW_CAVE | Average value for the range of available water capacity for the soil layer or horizon (cm of water per inches of soil depth) |

5  **Table C2. Adjusted $R^2$ (top) and p-values for Shapiro-Wilk test of normality of residuals (in parentheses below) for models predicting generalized Pareto parameters for HUCs 2 and 10 for POR- and MA-FDC**

| | HUC 2 | | HUC 10 | |
|---|---|---|---|---|
| | POR-FDC | MA-FDC | POR-FDC | MA-FDC |
| | 19% | 36% | 49% | 48% |
| Xi | (.005) | (.835) | (.502) | (.576) |
| | 98% | 97% | 91% | 88% |
| a | (.028) | (.103) | (.593) | (.593) |
| | 42% | 38% | 35% | 40% |
| k | (.602) | (.391) | (.080) | (.294) |

Regional regression results are given in tables C3-C6. These tables were produced with the help of the R package StarGazer (Hlavac, 2015).





**Table C3. Regional regression results for HUC 2 POR-FDC**

```
POR-FDC Regional Regression Results
===============================================================================
                      log(xi+1)           log(alpha)             log(k+1)
-------------------------------------------------------------------------------
log(DA_km2)              0.104               1.039
                      t = 2.873***        t = 41.058***
log(PPTAVG_BASIN)       0.643               2.338                 0.501
                      t = 1.385           t = 6.369***          t = 2.540**
log(WD_SITE)                                                      0.452
                                                               t = 3.187***
log(BFI_AVE)            0.548               0.204                 0.562
                      t = 1.861*          t = 0.725           t = 4.401***
log(AWCAVEcm)           0.399
                      t = 1.424
log(PET)                                   -2.672
                                         t = -8.132***
log(TOPWET)                                1.572
                                         t = 3.913***
Constant               -5.008               -3.277                -7.106
                      t = -2.091**        t = -1.321          t = -6.164***
-------------------------------------------------------------------------------
Observations            57                  57                    57
R2                      0.244               0.979                 0.450
Adjusted R2             0.186               0.977                 0.419
Residual Std. Error  0.271 (df = 52)     0.188 (df = 51)       0.117 (df = 53)
F Statistic       4.198*** (df = 4; 52) 466.896*** (df = 5; 51) 14.436*** (df = 3; 53)
===============================================================================
Note:                                      *p<0.1; **p<0.05; ***p<0.01
```

**Table C4. Regional regression results for HUC 2 MA-FDC**

```
MA-FDC Regional Regression Results
===============================================================================
                      log(xi+1)           log(alpha)             log(k+1)
-------------------------------------------------------------------------------
log(DA_km2)              0.156               1.036
                      t = 5.155***        t = 37.474***
log(PPTAVG_BASIN)       0.660               2.592                 0.479
                      t = 1.705*          t = 6.459***          t = 2.200**
log(WD_SITE)                                                      0.285
                                                               t = 1.821*
log(BFI_AVE)            0.612               0.031                 0.597
                      t = 2.492**         t = 0.100           t = 4.228***
log(AWCAVEcm)           0.290
                      t = 1.237
log(PET)                                   -2.606
                                         t = -7.258***
log(TOPWET)                                1.647
                                         t = 3.751***
Constant               -5.699               -4.449                -6.305
                      t = -2.852***       t = -1.641          t = -4.948***
-------------------------------------------------------------------------------
Observations            57                  57                    57
R2                      0.408               0.975                 0.410
Adjusted R2             0.362               0.972                 0.377
Residual Std. Error  0.226 (df = 52)     0.206 (df = 51)       0.129 (df = 53)
F Statistic       8.954*** (df = 4; 52) 394.099*** (df = 5; 51) 12.297*** (df = 3; 53)
===============================================================================
Note:                                      *p<0.1; **p<0.05; ***p<0.01
```





**Table C5. Regional regression results for HUC 10 POR-FDC**

```
POR-FDC Regional Regression Results
===========================================================================
                            log(xi+1)           log(alpha)          log(k+1)
---------------------------------------------------------------------------
log(DA_km2)                   0.445               1.171
                           t = 4.367***        t = 13.398***

log(PPTAVG_BASIN)             1.300               2.168              -0.028
                           t = 1.646           t = 4.235***       t = -0.076

log(WD_SITE)                                                        -0.590
                                                                   t = -1.632

log(BFI_AVE)                  1.620              -0.438              0.894
                           t = 2.508**         t = -0.758         t = 3.322***

log(AWCAVEcm)                 0.691
                           t = 1.157

log(PET)                                         -1.247
                                               t = -1.560

log(TOPWET)                                      -3.548
                                               t = -1.693

Constant                    -13.596              1.611              -1.654
                           t = -2.688**        t = 0.241          t = -0.787
---------------------------------------------------------------------------
Observations                   27                 27                 27
R2                            0.569              0.926              0.428
Adjusted R2                   0.491              0.908              0.353
Residual Std. Error     0.639 (df = 22)     0.464 (df = 21)     0.354 (df = 23)
F Statistic          7.259*** (df = 4; 22) 52.455*** (df = 5; 21) 5.736*** (df = 3; 23)
===========================================================================
Note:                                          *p<0.1; **p<0.05; ***p<0.01
```

**Table C6. Regional regression results for HUC 10 MA-FDC**

```
MA-FDC Regional Regression Results
===========================================================================
                            log(xi+1)           log(alpha)          log(k+1)
---------------------------------------------------------------------------
log(DA_km2)                   0.458               1.190
                           t = 4.403***        t = 11.352***

log(PPTAVG_BASIN)             1.370               2.188              -0.193
                           t = 1.699           t = 3.562***       t = -0.557

log(WD_SITE)                                                        -0.586
                                                                   t = -1.734*

log(BFI_AVE)                  1.614              -0.673              0.807
                           t = 2.446**         t = -0.970         t = 3.212***

log(AWCAVEcm)                 0.787
                           t = 1.291

log(PET)                                         -1.572
                                               t = -1.641

log(TOPWET)                                      -3.842
                                               t = -1.529

Constant                    -13.796              4.972              -0.555
                           t = -2.672**        t = 0.620          t = -0.283
---------------------------------------------------------------------------
Observations                   27                 27                 27
R2                            0.562              0.901              0.464
Adjusted R2                   0.482              0.877              0.395
Residual Std. Error     0.653 (df = 22)     0.557 (df = 21)     0.331 (df = 23)
F Statistic          7.057*** (df = 4; 22) 38.135*** (df = 5; 21) 6.650*** (df = 3; 23)
===========================================================================
Note:                                          *p<0.1; **p<0.05; ***p<0.01
```



**Acknowledgements**

Thanks to William Farmer for technical advice and to Jory Hecht for comments on an early draft. This material is based upon work supported by the National Science Foundation (NSF) Graduate Research Fellowship Program under Grant number DGE-1144081 and NSF Grant

number EEC-1444926. Any use of trade, firm, or product names is for descriptive purposes only and does not imply endorsement by the U.S. Government.

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

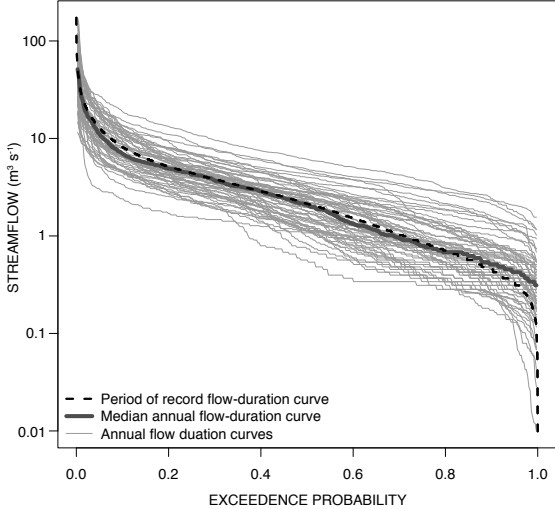

**Figure 1. Examples of period of record, median annual, and annual flow-duration curves for the Choptank River near Greensboro, Maryland, USA.**





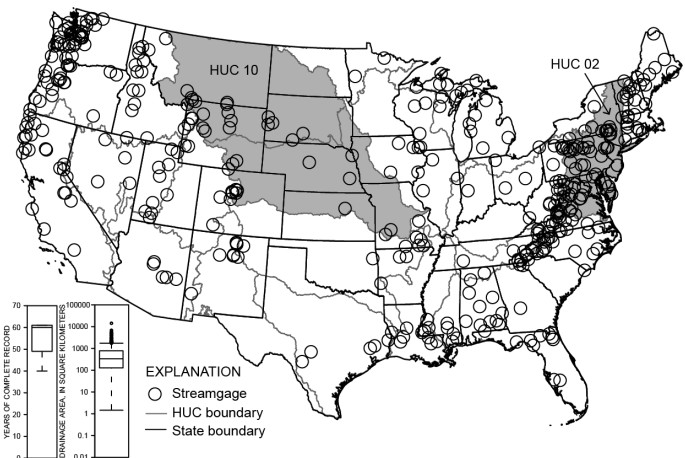

**Figure 2. Map of the conterminous United States with locations of the 420 stations shown in black and the two regions of regionalization shown in grey (HUC #10, Missouri region and HUC #02, the mid-Atlantic region.) Boxplots of the number of years of record length and the range of drainage areas are given in the lower left corner of the figure.**

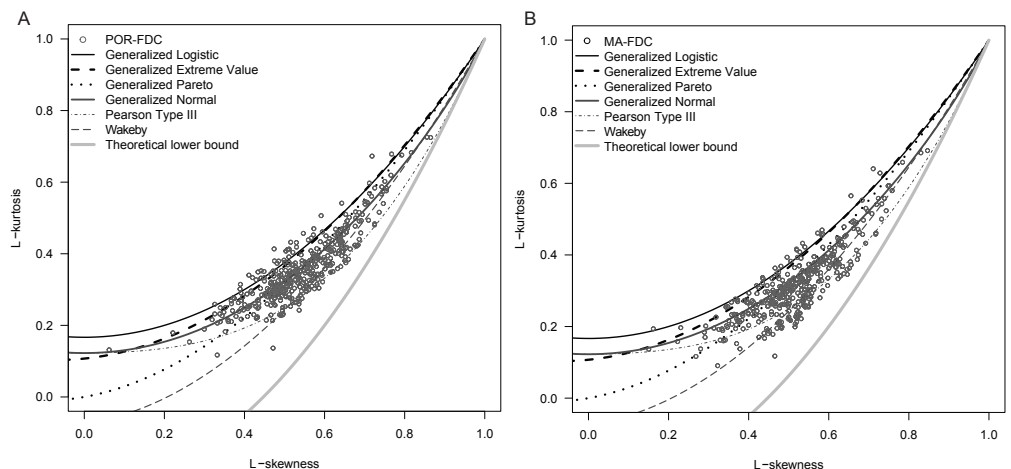

**Figure 3. L-Kurtosis ratios compared to L-skew ratios for POR-FDC (A) and MA-FDC (B) at 420 US sites as compared to the theoretical relations between L-Kurtosis and L-skew ratios for candidate probability distributions.**





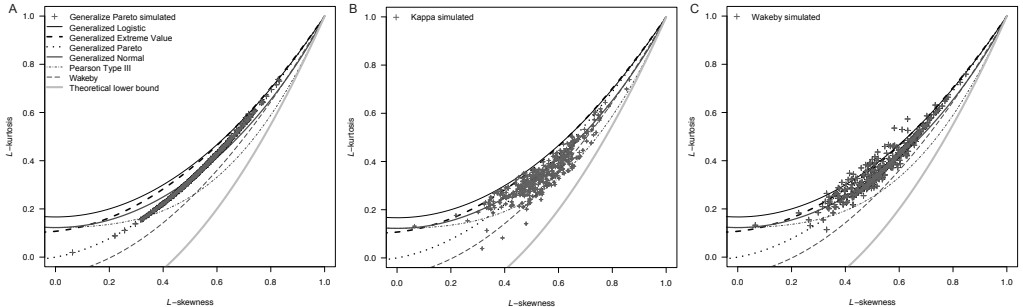

**Figure 4. L-Kurtosis ratios compared to L-skew ratios for POR-FDC simulated from sample L-moments estimates at 420 US sites using the 3-parameter generalized Pareto (A), the 4-parameter Kappa (B) and the 5-parameter Wakeby (C) distributions as compared to the theoretical relations between L-Kurtosis and L-skew ratios for candidate probability distributions.**

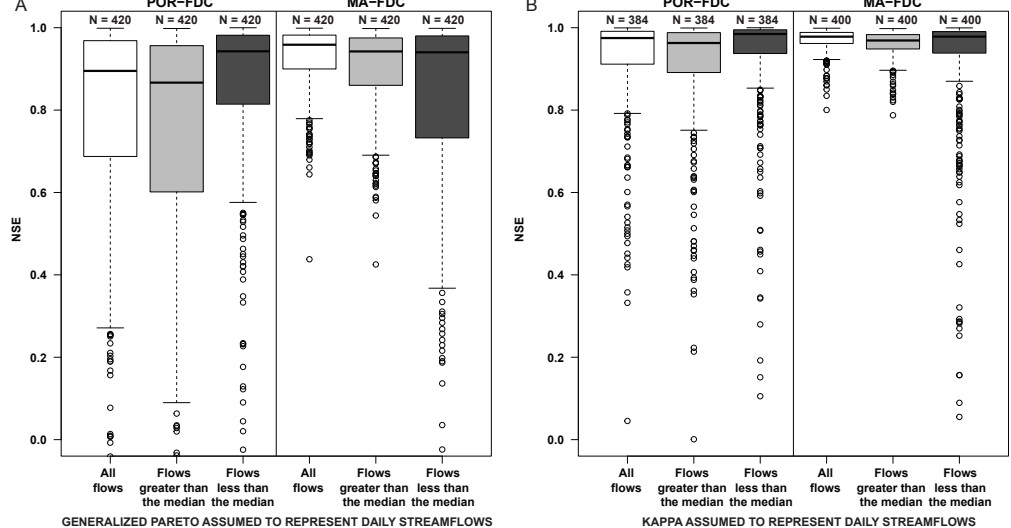

**Figure 5. Boxplots illustrating the range of Nash-Sutcliffe efficiency (NSE) values of period of record (POR-FDC) and median annual flow duration curves (MA-FDC) fit with the 3-parameter generalized Pareto distribution (A) and the 4-parameter Kappa distribution (B). Uneven sample sizes are due to sites for which Kappa parameters could not be estimated. To enable visual comparison, outliers below zero have been omitted. The number of negative NSE values outliers is (from left to right): 34,41,34,0,0,44, 0,0,1,0,0,1, respectively.**





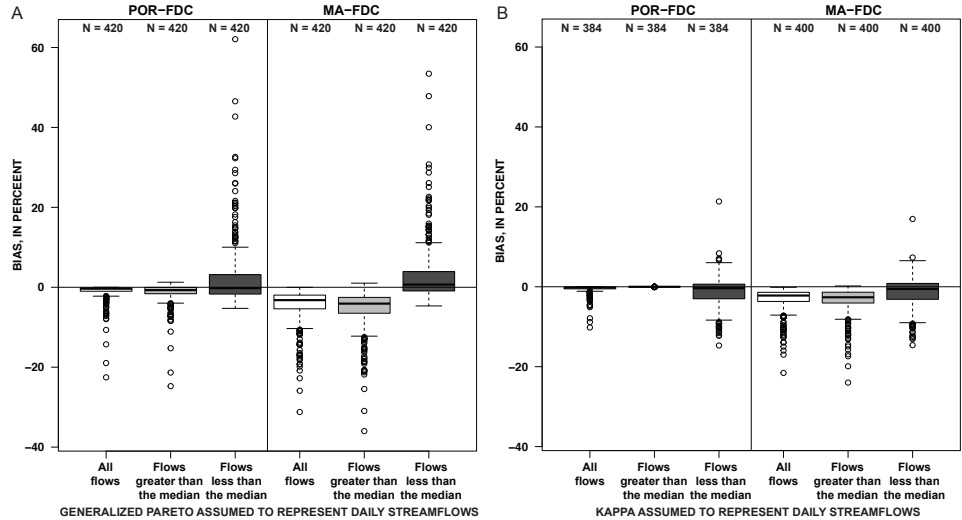

**Figure 6. Boxplots illustrating the range of bias, in percent, of period of record (POR-FDC) and median annual flow duration curves (MA-FDC) fit with the 3-parameter generalized Pareto (A) distribution and the 4-parameter Kappa distribution (B).**

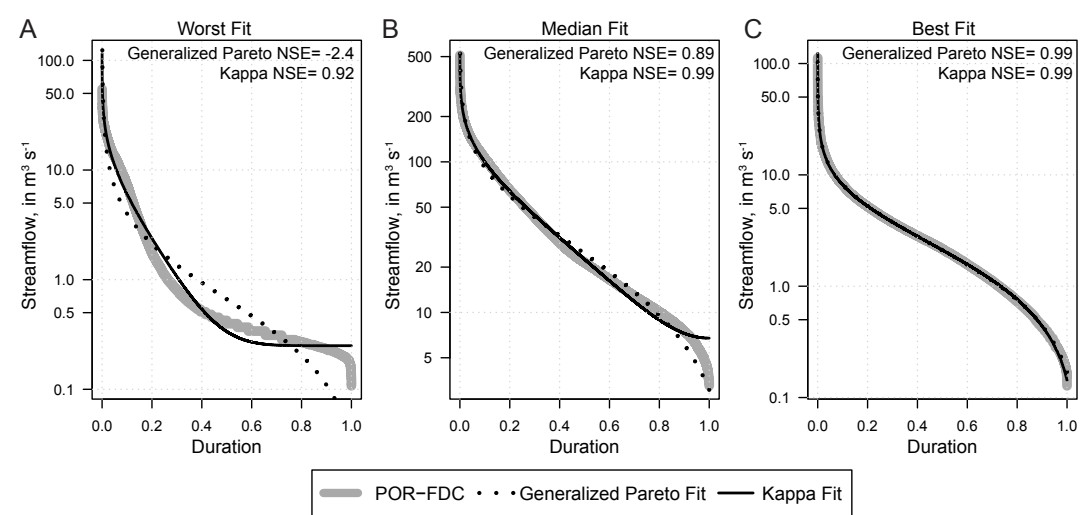

**Figure 7. Empirical period of record (POR) flow duration curves (FDCs) and models of FDCs with the 4-parameter Kappa distribution and the 3-parameter Generalized Pareto distribution for sites with the lowest (A), median (B), and highest (C) Nash Sutcliffe efficiency (NSE).**





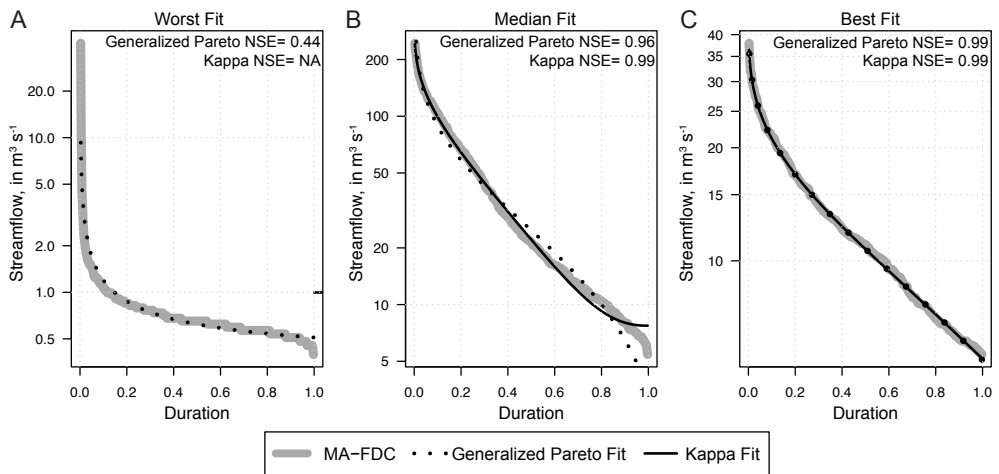

**Figure 8. Median annual flow duration curves (MA-FDCs) and models of FDCs with the 4-parameter Kappa distribution and the 3-parameter Generalized Pareto distribution for sites with the lowest (A), median (B), and highest (C) Nash Sutcliffe efficiency (NSE).**

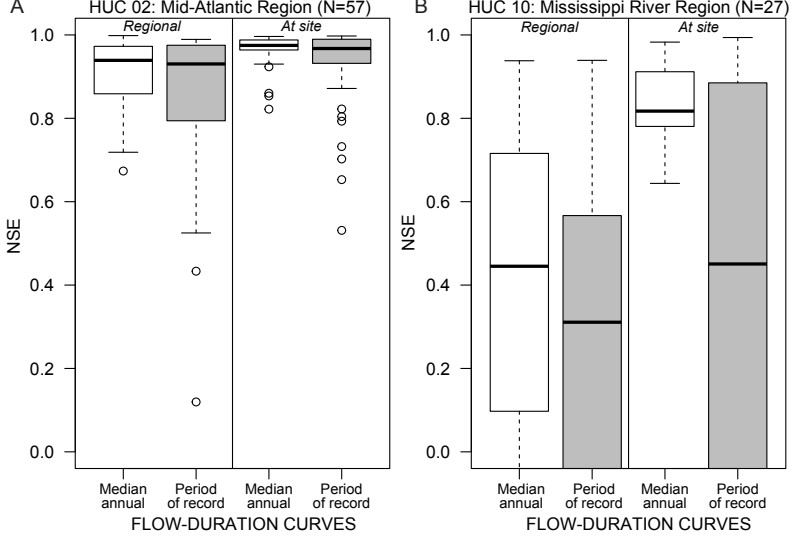

**Figure 9. Comparison of Nash-Sutcliffe efficiency (NSE) values for HUC 02 and HUC 10 regional models of POR-FDC and MA-FDC using leave-one-out cross validation. Boxplot comparisons of regional and at site models are**




**illustrated for MA-FDCs (white) and POR-FDCs (grey) for the HUC 02 (A) and HUC 10 (B) regions. To enable visual comparison, NSE coefficients below zero have been omitted. [For HUC 02, the number of outliers for the boxes left to right is 2,1,0,0; for HUC 10 left to right is 6,10,0.]**

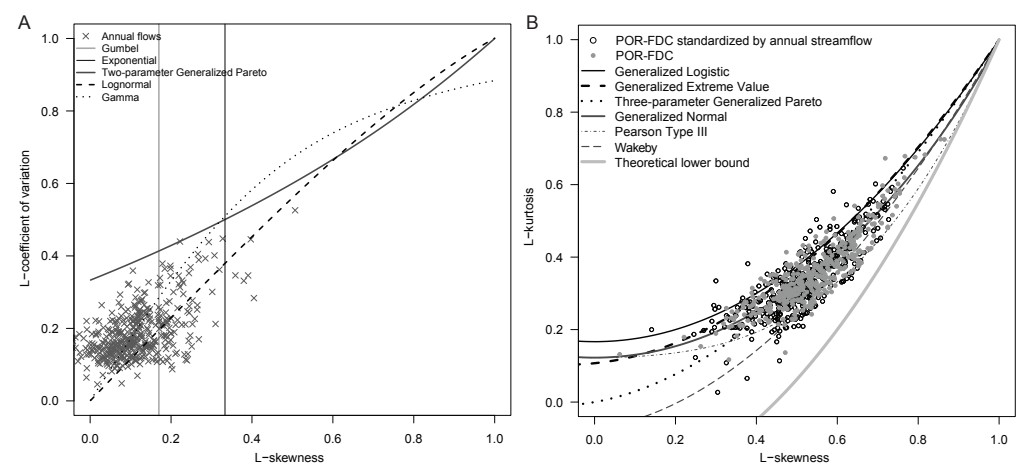

**Figure A1. Sample L-coefficient of variation (L-CV) ratios compared to sample L-skewness ratios for annual flows (A) and sample L-Kurtosis ratios compared to sample L-skewness ratios for POR-FDC standardized by annual flows (B) at 420 US sites. The theoretical relations between L-CV and L-skewness are shown for candidate two-parameter distributions in panel A; the theoretical relations between L-kurtosis and L-skewness are shown for candidate three-, four- and five-parameter distributions in panel B.**

