# Peer review of "On the probability distribution of daily streamflow in the United States"

_Hydrology and Earth System Sciences, 2016_

## Referee Comment (RC1) · Anonymous Referee #1 · 18 Oct 2016

The paper fits theoretical distributions to a large dataset of empirical streamflow observations covering the conterminous US. The study finds that median annual flow duration curves (FDC), which portray flow distribution in a typical year, can be reasonably fitted to three-parameter distributions. In contrast, period of record FDC that incorporate extreme streamflow variations over numerous years cannot be appropriately fitted, even to more complex theoretical distributions. The authors explore the implications of that finding on predictions in ungauged catchments using linear regressions in case studies. Predicting streamflow signatures (particularly FDC) in ungauged basins is both extremely useful and challenging and the findings of this study are interesting and important, particularly the insight that mAFDC might be both easier to predict and

more practically relevant than PoRFDC. However, there are two points that I would like to see further discussed before publication, as well as the few minor comments listed below.

First, the study is an impressive effort to fit FDCs to a very large dataset of unregulated catchments – this is definitely a key contribution of the paper. However, by covering the whole conterminous US, the dataset covers a wide variety of climates, catchment characteristics and flow regimes, and it would have been interesting to explore how the fit to specific distributions varies regionally. The shape of FDCs depicts the local flow regime, which are themselves related to climate and catchment characteristics (see e.g., Botter 2013). It would be nice to see whether there is a link between flow regimes, climate/catchment characteristics and the best fitted theoretical distribution. It would also be nice to discuss how the best-fit distributions relate to the distributions that might be expected from process-based models (Botter 2007, Botter 2009, Muller 2014, Muneepeerakul 2010, etc), given the dominant flow processes in particular catchments.

Second, while I appreciate the effort to extend an already complex and large scale analysis to prediction in ungauged basins, I would like to see more details on how the regression models were obtained (i.e. how the regression covariates were selected for Eqn 7-9), and a discussion on whether these regression models have a physical interpretation. Specifically, I am concerned about using linear regressions to estimate distribution parameters, which arguably have a more ambiguous physical interpretation as moments. Mean flow (first moment) for instance can be argued to be a linear combination of observable characteristics like mean rainfall, as per the water balance equation. The issue in regressing GPA3 parameters is that they are not linear combinations of the moments of the distribution, so using linear regressions to estimate the parameters does not allow moments to be linearly related. In other words, in this specific case, linearly regressed parameters are not compatible with a linear water balance relation on mean flow. To address this issue, please either apply linear regressions on

the moments of the distributions instead of the parameters (and discuss the physical interpretation of the linear models when appropriate), or make the case that Eqn 7-9 are not incompatible with water balance principles.

[To illustrate my point on linear regressions, let's assume the simplest linear model possible, where predictions are simply taken as the mean of the observed sample (this can happen in the specific case of the water balance model above if all catchments have an identical mean rainfall). Let's say that we have a sample of three catchments with the following GPA3 parameters and mean flow (computed from the parameters):

Basin | location param | scale param| shape param | Mean

1 | 0 | 100 | 0.3 | 143

2 | 0 | 1 |0.05 | 1

3 | 0 | 20 | 0.1 | 22

The predicted mean flow in a fourth catchment obtained from the observed mean flows (i.e. by taking the mean of the mean) is 47, whereas the mean flow computed from predicted GPA3 parameters (i.e. computed from the mean value of each parameter) is 55.]

Minor comments:

p5 l 17: please define GOF.

p12 l14-19: I have seen this issue most often addressed by taking the logarithm of flow quantiles before computing the NSE. Is there are reason why you preferred the selected approach?

p14 l13-15: I agree that modelling errors on FDCs are best assessed graphically and appreciate the effort of showing fits for particular basins with low, median and high NSE. Error duration curves (e.g., Muller 2016) are great way of visualizing performance fits over large samples (as opposed to individual basins), and it would be informative in my

opinion to display the relevant EFDCs for the whole dataset.

p.16 l3-4: I realize that concerns of overfitting regression models are to some extent addressed in the LOO cross validation analysis, but please display covariates statistics (e.g., quartiles and range or boxplot) to show that there is enough variability in the samples to credibly argue that the LOO performance is externally valid.

p.17 l2-4: Have you tested for serial correlation? Serial correlation affects the estimation of OLS standard errors and are particularly likely to occur between flow-connected gauges (i.e. gauges located on the same river).

p.18 l.11-20: The paper makes the great case that MAFDCs are easier to fit and have more practical relevance than PoRFDC. However, I would appreciate a more complete discussion of the tradeoff involved: mAFDC loses information on inter-annual variability, hence their better fit to "simpler" distributions. This is an important caveat that has strong implications for practical applications and should be made clearer in the discussion/conclusion in my opinion.

Table C1: I have trouble understanding how the BFI_AVE is a regression covariate for prediction in ungauged basins. My understanding is that flow observations are necessary to compute the BFI in the first place.

References:

Botter, G., Basso, S., Rodriguez-Iturbe, I., and Rinaldo, A.: Resilience of river flow regimes, Proceedings of the National Academy of Sciences, 110, doi: 10.1073/pnas.1311920 110, 2013.

Botter, G., Porporato, A., Rodriguez-Iturbe, I., and Rinaldo, A.: Basin-scale soil moisture dynamics and the probabilistic characterization of carrier hydrologic flows: Slow, leaching prone components of the hydrologic response, Water Resources Res, 43, doi:10.1029/2006WR005 043, 2007.

Botter, G., Porporato, A., Rodriguez-Iturbe, I., and Rinaldo, A.: Nonlinear storagedischarge relations and catchment streamflow regimes, Water resources research, 45, doi: 10.1029/2008WR007 658, 2009.

Müller, M. F., Dralle, D. N., and Thompson, S. E.: Analytical model for flow duration curves in seasonally dry climates, Water Resources Research, 50, doi:10.1002/2014WR015 301, 2014. Muneepeerakul, R., Azaele, S., Botter, G., Rinaldo, A., and Rodriguez-Iturbe, I.: Daily streamflow analysis based on a two-scaled gamma pulse model, Water Resources Research, 46, doi: 10.1029/2010WR009 286, 2010. Müller, M. F. and Thompson SE.: Comparing statistical and process-based flow duration curve models in ungauged basins and changing rain regimes, Hydrology and Earth System Sciences 20.2 669-683, 2016

---

## Referee Comment (RC2) · F. Serinaldi (Referee) · 18 Oct 2016

**General comments**

In this paper, the Authors perform a large scale analysis in order to identify a parametric distribution function providing reasonable approximation for flow duration curves (FDCs) across the conterminous United States. The paper relies on classical "weapons" in the "statistical" arsenal commonly applied in hydrology (L-moments, Nash-Sutcliffe performance index, linear regression in logarithmic space for regionalization, etc.). So, taking for granted that such tools are sound and correctly applied, the interest in this paper is not surely methodological, but concerns the empirical results. Considering that downloading data and analyzing them with R packages such as `lmom`, `lmomco` and some build-in regression functions is a matter of few hours (most of which are needed to slightly customize the default diagrams yielded by R), in my opinion, it is a bit hard to classify this kind of works as research papers. My very personal opinion, is that they can be at most technical reports or case studies (likely resulting from some master thesis). Anyway, I leave the paper classification to the Editors; from my side, I can only say that I cannot see significant insights, while there are some inconsistencies resulting in misleading conclusions. Just to make an example, there are works providing L-moments for gridded rainfall worldwide with quite limited insight about the nature of rainfall (e.g., Maeda et al., 2013), while others (e.g. Papalexiou and Koutsoyiannis, 2016) make similar analysis on gauged data but considering distribution families derived by entropy maximization, introducing a new test for seasonal variation, and providing a number of new insights. What I mean is that we can analyze a large data set "passively", by running e.g. R codes quite blindly, or we can decide to use data in order to understand the underlying processes in more depth. Said that, if we accept the first approach, the paper is ready to publish, once removed some nonsense discussed below (concerning the comparison of MA-FDC and POR-FDC); in the second case, we are far from a good quality work. In any case, I would like to use this opportunity to share my point of view on flow duration curves (FDC), stressing that the philosophy behind them probably needs some rethinking.

**Specific comments**

The Authors stress twice in the text that FDC (actually POR-FDC) "ignore the important serial stochastic structure of daily flows, including such issues as autocorrelation and seasonality" and they also recognize that simple 3- or 4-parameter distributions can only approximate FDCs. These statements are given in passing, but they are actually the core of the problem. In principle we can get whatever time series of numerical values, arranging it in ascending (descending) order and then plotting the sorted values against their rescaled ranks. Irrespective of the nature of the (numerical) data, the

result is always a monotonic pattern describing the function $g : \mathcal{R} \to [0, 1]$ (of course, the domain can be a subset of $\mathcal{R}$, and the function is strictly monotonic if there are not statistical ties (i.e. identical values)). If the aim is to fit a simple analytical function to such curves, theoretical cumulative distribution functions (CDFs) seem to be natural candidates. However, CDFs are not simple curves useful for fitting data, but represent the nonexceedance probability of a random variable and work if the data are independent and identically distributed (*iid*). All these concepts are trivial and the Authors know them better than me. However, since daily stream flow records surely do not fulfill any of these conditions, why should a single distribution fit FDCs? In other words, in spite of the efforts made along the years to find suitable CDFs for modeling FDCs, the problem is ill-posed by definition: even the most parametrized CDFs cannot mimic FDCs unless the flow series is characterized by strong mixing (e.g. weak seasonal pattern compared to non-seasonal (essentially "random") fluctuations). So, if the FDCs analysis reduces to a simple exercise of curve fitting, the overall analysis performed in this type of studies can make sense; otherwise, if the aim is to fit a CDF, and then concluding that such model describe probability of (non)exceedance or something like that, this statement can be much more problematic, unless the model is a mixture of CDFs describing data approximately 'identically distributed' (id) such as seasonal or monthly subsets. In fact, the analysis reported by e.g. Basso et al. (2015) is performed on a seasonal basis.

More generally, as the Authors know, stream flows are characterized by two properties that play a fundamental role in this context: seasonality and persistence (often long range persistence; see e.g. Montanari et al. (1997,2000) or more recently Serinaldi and Kilsby (2015)). Seasonality is often the main source departure from *id* condition. This is well known for instance in rainfall modeling where simple 2-parameter Weibull distributions are surely insufficient to describe daily rainfall over the entire year, but their performance is very good if we introduce parameters varying with the seasonality. Indeed the fact that stream flow values can cover two or three orders of magnitude simply depends (obviously) from the alternation of high-flow and low-flow seasons, in which

the *id* hypothesis is far from being realistic. On the other hand, long-range dependence results in inter-annual variability, which is what the index-flood method attempts to take into account in quite a naïve way. However, the index-flood still overlooks the problem of non-*id* conditions within calendar or water year. When the seasonal signal is strong, this can be the main reason for the lack of fitting of simple parametric distributions, and index-flood cannot improve the fitting very much. Moreover, while seasonality impacts on the overall shape of flow distribution (imagine to mix e.g. 12 different distributions, each reproducing approximately *id* monthly flows), long range dependence induces inter-annual fluctuations that impact especially on the tails. Therefore, the index-flood method adjusts more easily tail behavior than the overall shape of the parametric FDC.

The above remarks, can help to understand how to improve FDC if we want to avoid physical approaches *à la* Botter (...but overlooking physical arguments is never a good choice) and keep the model purely statistical, but a little bit more coherent with the nature of the data. The easiest approach is surely splitting data at e.g. seasonal scale. On the other hand, we can build on the fact that the regionalization procedure commonly applied in hydrology (and summarized in this study) is only a rough and naïve version of generalized linear/additive models (GLM/GAM an their extensions) $f(y; \theta(\mathbf{X}))$, where $f$ is the distribution of flows $Y$, $\theta$ is a vector of parameters (e.g., the three parameters of the Generalized Pareto) and $\mathbf{X}$ is a design matrix of covariates (e.g., the variables in Eqs. 7-9). In this framework, seasonality can easily be introduced by simple sine and cosine functions describing the seasonal cycles; since a couple of waves are generally sufficient to describe the seasonal flow regime, GLMs imply only a couple of additional parameters. Alternatively, a factor index can be used in the fitting procedure to distinguish e.g. between the four seasons or the 12 months. In all cases, the resulting model not only account for the spatial variability but also for the non-id conditions by a few additional parameters that have a clear physical interpretation (they represent the seasonal regimes across the area of study). Of course, the usual graphical representation (as in Fig. 1) is possible only if we compare observations and simulations because such a diagrams merge quantiles coming from a set of distributions (devised

for $id$ data), roughly speaking one for each season (or month). However, this is not surprising because the observed FDCs themselves incorporate values coming from different (seasonal) distributions, thus explaining the lack of fiit of simple models. This approach also helps overcoming the problem of MA-FDC simulation mentioned in the paper. Notice that the effect of seasonal variation as well as long range dependence can be recognized in Figs. 7(a-b) and 8(b-c) in the form of multimodality, while the step-wise pattern in some regions of the FDCs in Fig. 7(a) and 8(a) denotes the presence of statistical ties, which generally results from limits in the resolution of measurement devices or round-off procedures. The first aspect denotes the intrinsic inadequacy of whatever classical unimodal distribution, while the latter often affects estimation procedures (so, I'm not so surprised about the poor fitting). In this respect I have to say that the scale of the x-axis does not help fitting assessment. I'm a bit surprised because after Vogel and Fennessey (1994), we know that stretched axes enhancing the linearity of FDCs and CDFs allow much better assessment, in agreement with recommendations available in the literature on visual perception and data visualization (see e.g., works by Tufte, Cleveland, etc.).

Another concern is about the comparison of POR-FDCs and MA-FDC. The Authors conclude that fitting MA-FDCs is easier and more reliable than POR-FDCs as "prediction of POR-FDCs was less consistent" (consistent?). The comparison between MA-FDCs and POR-FDCs is ill-posed by itself and in the interpretation of NSE. Firstly, for MA-FDC, we always fit a CDF on 365 data points, where each one is the median (or mean) of a set of $M$ values, where $M$ is the number of years (here 40-60); for POR-FDCs we are trying to fit a CDF on $365 \cdot M$ values (i.e. a sample 40-60 times larger), where each values (order statistics) should be the point estimates of the corresponding quantiles. In the first case, we seek the fitting in the range of probabilities $\left(\frac{1}{365+1} \approx 3 \cdot 10^{-3}, \frac{365}{365+1} \approx 0.997\right)$, whereas in the second we pretend to fit quantiles corresponding to probabilities between $\frac{1}{365M+1} \approx 5 \cdot 10^{-5}$ and $\frac{365M}{365M+1} \approx 0.99995$. So, is it so surprising that fitting a curve on 365 "smoothed" values (medians) is easier than

on 18250 values (being already aware that such values cannot come, by definition, from a unique distribution)?

Secondly, the above remark allows some reflection on the (mis)use of performance metrics and their interpretation. As for every performance index (absolute metrics, relative errors, deviance or similarity measures, information criteria, etc.), NSE (which is simply the similarity index corresponding to the mean squared error) is devised to compare the performance of a set of models for the **same** data set; in our case, not only the sample size of the data sets and error terms is completely different (365 against about 18250), but also the nature of the data is completely incomparable (raw data against medians resulting from a very specific selection procedure). Thus, stating that NSE for MA-FDC is generally smaller than that of POR-FDCs is nonsense, as we are comparing apples with pears. Moreover, even though I know that hydrologists have fallen in love with NSE for some esoteric reason, I would like to stress that a performance index should be chosen according to the particular type of discrepancy one wants to highlight, and not because it is popular. To be more specific, NSE is a similarity index comparing the errors from the selected model (numerator) with those from a benchmark or reference model (denominator), where the reference model is, in this case, the sample average (aka 'reference climatology' in climatological literature or "naïve" reference in forecasting literature...it seems that people in each discipline like renaming the same concepts many times, just to increment a little bit the already widespread confusion...). The choice of this "naïve" reference has two consequences: (1) the range of possible NSE values is strongly asymmetric, and (2) every model more complex than the simple average easily yields relatively high NSE values; this is usually interpreted as a good performance, but actually it is not, because the way NSE values populate the range $(-\infty, 1)$ is strongly nonlinear. Since the average is not a sufficient statistics even for data coming from a Gaussian distribution, it is easy to recognize that whatever model provides great improvement and (relatively high NSE) compared to such "naïve" reference. Therefore, sentences such as "Despite this comparable fit, the NSE coefficients are quite different: 0.89 for POR-FDC GPA3 versus the much

higher 0.96 for MA-FDC GPA3. This discrepancy reflects a challenge in the use of the metric and indicates why visual inspection of FDC plots is particularly important for understanding overall GOF", make little sense because (1) the two values refer to different data sets (comparisons can be done only between at-site and regional models for the same data set, MA and POR, respectively), and (2) even if they referred to different models for the same data set, NSE is not equipped with criteria allowing to say if the difference between two values is significant or not (unlike methods based on maximum likelihood and/or information criteria). Concerning the rationale, choice and interpretation of performance measures please see Dawson et al. (2007), Hyndman and Koehler (2006), Jachner et al. (2007), Burnham and Anderson (2004), Reusser et al. (2009), among others.

**Technical remarks**

Please use homogeneous notation: "2-,3-,4-parameter distributions" or "two-,three-,four-parameter distributions" throughout the text.

P3L16: it can be worth citing Doulatyari et al (2005), Basso et al. (2015), and Schaefli et al. (2013)

P6L10-15: the Authors refer to other quantile estimators; however, Weibull plotting position is not a quantile estimator. In this respect , it can also be worth having a look at Makkonen (2006), and Hutson (2000)

P7L8: "Hosking and Wallis 1997"

P7L16: "natural logarithm"

P7L16: "linear combination of order statistics" can better reflect their actual rationale (linear combination with weighted moments is a consequence)

P8L16: "see e.g. Rianna et al. (2011) and references therein"

P9L10-15: I may have missed something, but I cannot see where the effect of sample

size on L-moment scattering is shown. Moreover, the similarity between L-moments of POR-FDC and MA-FDC (Fig. 3) are likely due to the fact that L-moments are less sensitive to tail behavior by definition. Since the body of POR-FDC and MA-FDC are similar, L-moment ratios are similar.

P9L25: I understand the attitude of simulating everything, but sometimes it is not strictly necessary; in this case, we already known that daily stream flows cannot be distributed as GPA (or whatever else common unimodal distribution) because they are non-$iid$.

P11L5-25: concerning the simulation of MA-FDC, I think you can do an attempt by fitting the standardized annual sequences (dividing each year of data by annual median), then simulating from this distribution, and multiplying each simulated block of 365 elements by resampled (bootstrap) values of annual medians. This way, you do not need to fit any distribution for annual medians, but can explore the effect of inter-annual variability (see discussion above). Concerning the index-flow method, please note that my paper highlights a problem that is probably more serious than the uncertainty of the distribution of annual medians. Actually, the analysis of confidence intervals highlighted that the distribution of the product $Q \cdot X'$ is not coherent with the idea of constant median over blocks of 365 elements, because the distribution of the product of two random variables implies the product of independent realizations, whereas the annual (constant median) introduces redundancy. This is one of the reasons why I think that common FDC frameworks, even if simple, actually provide very rough approximations, and should be replaced by more coherent methods, even if this means to loose this (perhaps excessive) simplicity.

P14L5-10: see comment above about a more careful choice of graphical properties, axes scales, etc.

P16L1: which is the physical interpretation of GPA scale and shape?

P17L4: "Table C2"

Appendix A: I understand the rationale of this discussion, but the comparison between index-flood models and classical FDCs should be done in terms final output. As mentioned above, index-flood is an attempt to account for one of the key aspects of stream flows (inter-annual variability or long range fluctuations and persistence if you prefer). It suffers some statistical inconsistencies (that may be overcome by moving from the distribution of a product to e.g. compound distributions, but this needs to be explored) and does not account for the second and perhaps more important aspect, i.e. non-*id* intra-annual conditions. This is just to say that the problem goes slightly beyond distribution fitting, but requires a more careful consideration of the nature of data and underlying process, otherwise it reduces to what Vit Klemes called "dilettantism in hydrology" (i.e. replacing physics with (often misused) statistics).

Sincerely,

Francesco Serinaldi

**References**

Basso S, M Schirmer, G Botter (2015) On the emergence of heavy-tailed streamflow distributions Advances in Water Resources 82, 98-105

Burnham K. P. and Anderson D. R. (2004) Multimodel Inference: Understanding AIC and BIC in Model Selection, Sociological Methods Research 2004; 33; 261

Dawson, C.W., Abrahart, R.J., See, L.M., (2007). Hydrotest: a web-based toolbox of evaluation metrics for the standardised assessment of hydrological forecasts. Environ. Modell. Soft. 22, 1034–1052

Doulatyari B, A Betterle, S Basso, B Biswal, M Schirmer, G Botter (2015) Predicting streamflow distributions and flow duration curves from landscape and climate Advances in Water Resources 83, 285-298,4

Hutson, A.D. (2000) A composite quantile function estimator with applications in bootstrapping, Journal of Applied Statistics, 27 (2000), pp. 567–577

Hyndman, R.J., Koehler, A.B., (2006). Another look at measures of forecast accuracy. Int. J. Forecast. 22, 679–688

Jachner, S., van den Boogaart, K.G., Petzoldt, T., (2007). Statistical methods for the qualitative assessment of dynamic models with time delay. J. Stat. Softw. 22 (8), 1–30

Maeda, E. E., Arevalo Torres, J. and Carmona-Moreno, C. (2013), Characterisation of global precipitation frequency through the L-moments approach. Area, 45: 98–108. doi:10.1111/j.1475-4762.2012.01127.x

Makkonen L (2006), Plotting positions in extreme value analysis, Journal of Applied Meteorology and Climatology 45 (2), 334-340

Montanari, A.; Rosso, R.; Taqqu, M.S. (1997) Fractionally differenced ARIMA models applied to hydrologic time series: Identification, estimation, and simulation. Water Resour. Res., 33, 1035–1044.

Montanari, A.; Rosso, R.; Taqqu, M.S. (2000) A seasonal fractional ARIMA model applied to the Nile River monthly flows at Aswan. Water Resour. Res., 36, 1249–1259.

Papalexiou S.M., and D. Koutsoyiannis (2016), A global survey on the seasonal variation of the marginal distribution of daily precipitation, Advances in Water Resources, 94, 131–145, doi:10.1016/j.advwatres.2016.05.005.

Reusser, D.E., Blume, T., Schaefli, B., Zehe, E., (2009). Analysing the temporal dynamics of model performance for hydrological models. Hydrol. Earth Syst. Sci. 13, 999–1018.

Rianna, M., Russo, F., and Napolitano, F. (2011) Stochastic index model for intermittent regimes: from preliminary analysis to regionalisation, Nat. Hazards Earth Syst. Sci., 11, 1189-1203, doi:10.5194/nhess-11-1189-2011

Schaefli B, A Rinaldo, G Botter (2013) Analytic probability distributions for snow-
dominated streamflow Water Resources Research 49 (5), 2701-2713

Serinaldi F, Kilsby CG. (2016) Understanding persistence to avoid underestimation of collective flood risk. Water, 8(4), 152.

---

## Referee Comment (RC3) · Anonymous Referee #3 · 18 Oct 2016

In their paper Blum et al. applied some well-known methodologies for finding suitable probability distributions for both period-of-record (POR) and median annual (MA) Flow Duration Curves (FDCs) in a very large area, such as the conterminous US. The authors found that, for the huge number of gauges analyzed, both the 4-parameter kappa and 3-parameter generalized Pareto distributions can reasonably simulate MA-FDC, while on the contrary even more complex distributions are unable to fit completely the very complex behavior of POR-FDCs, which explicitly accounts for extreme values. Furthermore, the authors also provide an example on possible application of their results for predicting FDC in ungauged sites, by means of the linear regression technique.

While the paper does not present in my opinion any relevant novelty from the methodological point of view, the effort of the authors to fit FDCs to such a large dataset has to be underlined.

I have few minor comments about the manuscript, that I list below. I hope my comments can help to improve further the paper.

Since the research does not deal with intermittent streams, and a relevant percentage of sites (170 on 590, almost 30%) was not considered into the analysis, I would suggest to slightly modify the title of the contribution, in order to make it more fitting with the content. I suggest something like this: "The probability distribution of daily streamflow in the perennial rivers of conterminous United States". Furthermore, some words would be appreciated about future research concerning intermittent streams in conterminous US.

Paragraph 3.1 and Figure 3: due to the huge extension of the study area and the number of catchments analyzed, it would be interesting to verify if specific distributions fit better to specific regions or other climate/catchment features. I suggest to go at least a bit into details with this point. For example (but it's just an idea) points in Figures 3A and 3B can have different colors depending on different regions (and/or other climate/catchment distinctive features).

P 11 l 18-20: I would rather say that "the selection [. . .] may be as challenging as [. . .]". However, among the theoretical advantages associated to the index flow method, there is the fact that complexity of Kappa and GPA distributions applied to the dimensionless daily streamflow is reduced, since the parameter alpha can be achieved as a combination of the other distribution parameters (please refer to Castellarin et al., 2007). This is a very important feature for regionalization studies. I would include this comment in the discussion

P 16 Eq. 7: I'm confused about using BFI as an explanatory variable, since to my knowledge it should be calculated/estimated from observed/estimated streamflow. Perhaps this variable can be replaced by some others accounting for the influence of litho-

logical features on streamflows

P 18 l 6-14: I acknowledge limitations and drawbacks of using POR-FDCs, but the discussion seems to me too 'biased' towards MA-FDCs. I suggest a more detailed discussion, so that also the final sentence (l 19: "MA-FDCs [. . .] should not be used when severe floods and droughts are of interest") is better contextualized.

Finally, please consider to edit the text following the suggested corrections:

P 4 l 12: "When additional goodness-of-fit (GOF) metrics. . ." so that you can use the acronym later (from P 5 l 17 onwards)

Paragraphs 2.1 and 2.2: please correct the numbering

P 6 l 20: "Figure 6 illustrates the differences. . ."

P 7 l 16: I think that the sentence "where log represents the natural log" should be moved to line 11.

P 11 l 3: maybe it could be useful for the reader if authors comment a little bit more the figure, highlighting briefly why L-moment ratios simulated from WAK are less consistent than those simulated from KAP.

Captions Fig. 7 and Fig. 8: it is useful to highlight that lowest, median and highest NSE values are referred to GPA probability.

Figure 9 caption: I guess one number is missing concerning the number of outliers for HUC 10

---

## Short Comment (SC1) · 13 Dec 2016

Anonymous Referee 1 The paper fits theoretical distributions to a large dataset of empirical streamflow observations covering the conterminous US. The study finds that median annual flow duration curves (FDC), which portray flow distribution in a typical year, can be reason- ably fitted to three-parameter distributions. In contrast, period of record FDC that in- corporate extreme streamflow variations over numerous years cannot be appropriately fitted, even to more complex theoretical distributions. The authors explore the implica- tions of that finding on predictions in ungauged catchments using linear regressions in case studies. Predicting streamflow signatures (particularly FDC) in ungauged basins is both extremely useful and challenging and the findings of this

study are interesting and important, particularly the insight that MAFDC might be both easier to predict and more practically relevant than PoRFDC. However, there are two points that I would like to see further discussed before publication, as well as the few minor comments listed below.

First, the study is an impressive effort to fit FDCs to a very large dataset of unregulated catchments – this is definitely a key contribution of the paper. However, by covering the whole conterminous US, the dataset covers a wide variety of climates, catchment characteristics and flow regimes, and it would have been interesting to explore how the fit to specific distributions varies regionally. The shape of FDCs depicts the local flow regime, which are themselves related to climate and catchment characteristics (see e.g., Botter 2013). It would be nice to see whether there is a link between flow regimes, climate/catchment characteristics and the best fitted theoretical distribution. It would also be nice to discuss how the best-fit distributions relate to the distribu- tions that might be expected from process-based models (Botter 2007, Botter 2009, Muller 2014, Muneepeerakul 2010, etc), given the dominant flow processes in particu- lar catchments.

RESPONSE: We appreciate these excellent suggestions. In the revised manuscript, we will add an analysis that assesses the fit of the FDCs within each of 19 major hydrologic regions of the United States to supplement the nationwide results. We also intend to evaluate how the goodness of fit of FDC models relate to various climate and catchment characteristics to enable a better understanding of those situations which are most challenging to characterize FDCs.

Second, while I appreciate the effort to extend an already complex and large scale analysis to prediction in ungauged basins, I would like to see more details on how the regression models were obtained (i.e. how the regression covariates were selected for Eqn 7-9), and a discussion on whether these regression models have a physical interpretation. Specifically, I am concerned about using linear regressions to estimate distribution parameters, which arguably have a more ambiguous physical interpretation as moments. Mean flow (first moment) for instance can be argued to be a linear combination of observable characteristics like mean rainfall, as per the water balance equation. The issue in regressing GPA3 parameters is that they are not linear combinations of the moments of the distribution, so using linear regressions to estimate the parameters does not allow moments to be linearly related. In other words, in this specific case, linearly regressed parameters are not compatible with a linear water balance relation on mean flow. To address this issue, please either apply linear regressions on the moments of the distributions instead of the parameters (and discuss the physical interpretation of the linear models when appropriate), or make the case that Eqn 7-9 are not incompatible with water balance principles. [To illustrate my point on linear regressions, let's assume the simplest linear model possible, where predictions are simply taken as the mean of the observed sample (this can happen in the specific case of the water balance model above if all catchments have an identical mean rainfall). Let's say that we have a sample of three catchments with the following GPA3 parameters and mean flow (computed from the parameters): Basin | location param | scale param| shape param | Mean 1 | 0 | 100 | 0.3 | 143 2 | 0 | 1 |0.05 | 1 3 | 0 | 20 | 0.1 | 22 The predicted mean flow in a fourth catchment obtained from the observed mean flows (i.e. by taking the mean of the mean) is 47, whereas the mean flow computed from predicted GPA3 parameters (i.e. computed from the mean value of each parameter) is 55.]

RESPONSE: We agree that assuming that the GPA model parameters are independent of each other is a poor assumption. Based on this comment, as well as several comments below along with concerns raised by other reviewers, we have decided to remove this case study from the revised manuscript and add to the manuscript a deeper exploration of the regional and seasonal behavior of the goodness of fit of FDC models, in addition to our national analysis.

Minor comments: p5 l 17: please define GOF.

RESPONSE:We will add this definition.

p12 l14-19: I have seen this issue most often addressed by taking the logarithm of flow quantiles before computing the NSE. Is there are reason why you preferred the selected approach?

RESPONSE:We agree that taking the logarithm of the flow quantiles before computing the NSE is a preferred method. We had originally employed NSE in real space due to the occurrence of zero flow values and, therefore, we could not take logarithms of the flows. However, we subsequently removed those streamgages with zero flows, but kept the original reporting of the goodness of fit. Thus, your point is valuable and in the revised manuscript, we will report the NSE values of the logarithms of the streamflow.

p14 l13-15: I agree that modelling errors on FDCs are best assessed graphically and appreciate the effort of showing fits for particular basins with low, median and high NSE. Error duration curves (e.g., Muller 2016) are great way of visualizing performance fits over large samples (as opposed to individual basins), and it would be informative in my opinion to display the relevant EFDCs for the whole dataset.

RESPONSE: Thank you for this useful comments. We will produce this graphic in our revisioins and explore the value of adding it to the manuscript.

p.16 l3-4: I realize that concerns of overfitting regression models are to some extent addressed in the LOO cross validation analysis, but please display covariates statistics (e.g., quartiles and range or boxplot) to show that there is enough variability in the samples to credibly argue that the LOO performance is externally valid.

RESPONSE:This is a good point, however this Leave One Out Analysis will be removed along with the case study.

p.17 l2-4: Have you tested for serial correlation? Serial correlation affects the estimation of OLS standard errors and are particularly likely to occur between flow-connected gauges (i.e. gauges located on the same river).

We thank the author for pointing out this question; we had not examined if there are

nested basins in our dataset that may affect the OLS standard errors. Nevertheless, we are removing the case study now from the analysis.

p.18 l.11-20: The paper makes the great case that MAFDCs are easier to fit and have more practical relevance than PoRFDC. However, I would appreciate a more complete discussion of the tradeoff involved: mAFDC loses information on inter-annual variability, hence their better fit to "simpler" distributions. This is an important caveat that has strong implications for practical applications and should be made clearer in the discussion/conclusion in my opinion.

RESPONSE:We will emphasize this important distinction between MAFDC's and PoRFDC's in the revised manuscript.

Table C1: I have trouble understanding how the BFI_AVE is a regression covariate for prediction in ungauged basins. My understanding is that flow observations are necessary to compute the BFI in the first place.

RESPONSE:The BFI_AVE is available as a grid generated by inverse-distance-weighting of base-flow index values computed at USGS streamgages for the entire United States (Falcone et al., 2010). Nevertheless, we are removing the case study from the analysis.

Falcone, J. A., D. M. Carlisle, D. M. Wolock, and M. R. Meador (2010), GAGES: A stream gage database for evaluating natural and altered flow conditions in the conterminous United States, Ecology, 91(2), 621–621, doi:10.1890/09-0889.1.

---

## Short Comment (SC2) · 13 Dec 2016

General comments In this paper, the Authors perform a large scale analysis in order to identify a parametric distribution function providing reasonable approximation for flow duration curves (FDCs) across the conterminous United States. The paper relies on classical "weapons" in the "statistical" arsenal commonly applied in hydrology (L-moments, Nash-Sutcliffe performance index, linear regression in logarithmic space for regionalization, etc.). So, taking for granted that such tools are sound and correctly applied, the interest in this paper is not surely methodological, but concerns the empirical results. Considering that downloading data and analyzing them with R packages such as lmom, lmomco and some build-in regression functions is a matter of few hours

(most of which are needed to slightly customize the default diagrams yielded by R), in my opinion, it is a bit hard to classify this kind of works as research papers. My very personal opinion, is that they can be at most technical reports or case studies (likely resulting from some master thesis).

RESPONSE: The reviewer's comments are very far from the truth concerning the effort involved in an empirical study of this type. The 'empirical analysis' which was performed in this paper began as empirical analyses in Figures 2 and 3 in the paper by Vogel and Fennessey (1993) which only examined the pdf of daily streamflow for 23 sites in Massachusetts. To date, the paper by Vogel and Fennessey (1993) has 346 google scholar citations, and arose entirely from the empirical challenge described in the manuscript under consideration by HESS. That empirical work was later extended by Archfield (2009) to the New England region, but was never published. Furthermore, the process of fitting a probability distribution to daily streamflow observations led to numerous challenges associated with zero occurrences, and obtaining plausible estimates of the lower bound of daily streamflow. It is only through such empirical studies that researchers can become aware of many of the very practical (and perhaps mundane to the reviewer) challenges and concerns in hydrology.

Anyway, I leave the paper classification to the Editors; from my side, I can only say that I cannot see significant insights, while there are some inconsistencies resulting in misleading conclusions. Just to make an example, there are works providing L-moments for gridded rainfall worldwide with quite limited insight about the nature of rainfall (e.g., Maeda et al., 2013), while others (e.g. Papalexiou and Koutsoyiannis, 2016) make similar analysis on gauged data but considering distribution families derived by entropy maximization, introducing a new test for seasonal variation, and providing a number of new insights. What I mean is that we can analyze a large data set "passively", by running e.g. R codes quite blindly, or we can decide to use data in order to understand the underlying processes in more depth. Said that, if we accept the first approach, the paper is ready to publish, once removed some nonsense discussed below (concerning the comparison of MA-FDC and POR-FDC); in the second case, we are far from a good quality work. In any case, I would like to use this opportunity to share my point of view on flow duration curves (FDC), stressing that the philosophy behind them probably needs some rethinking.

RESPONSE: We welcome the detailed and insightful comments on our work, however, we note a tremendous gap between the personal thinking on this problem of this reviewer and the vast literature on FDC's.

RESPONSE: In spite of the fact that daily streamflows are certainly not independent, and probably not identically distributed, FDC's have found widespread use in practice, and numerous studies have shown that the assumption of a fixed pdf for daily streamflow can lead to improvements in our ability to estimate streamflows at ungaged sites and numerous other applications. Thus, as a practical matter, in spite of the fact that daily streamflows may not really arise from an iid process, there are many advantages of making the empirical assumption of an identically distributed process.

RESPONSE: We encourage theoretical work by the reviewer and others to explore the possibility of modeling the probability distribution of daily streamflow as a mixture of several pdfs, however we remain quite confident that our empirical explorations of the distribution of daily streamflow may be quite useful in practice, particularly for the case of prediction in ungauged basins. It is this motivation - prediction at ungaged basins - that motivated this analysis, as the search for parsimonious solutions to fundamental problems in hydrology will always remain of use in our field. Furthermore, we note that this particular paper is our first attempt to summarize more recent investigations mentioned above which are the result of several decades of research on this topic. Thus we take great exception to numerous comments of Serinaldi when he appears to denigrate the empirical nature of work on FDC's.

RESPONSE: We appreciate the recommendation of the author concerning Papalexiou and Koutsoyiannis (2016) which serves to further motivate the importance and relevance of our work. We will add a discussion of Papalexiou and Koutsoyiannis (2016) to our revised paper.

Specific comments

The Authors stress twice in the text that FDC (actually POR-FDC) "ignore the important serial stochastic structure of daily flows, including such issues as autocorrelation and seasonality" and they also recognize that simple 3- or 4-parameter distributions can only approximate FDCs. These statements are given in passing, but they are actually the core of the problem. In principle we can get whatever time series of numerical values, arranging it in ascending (descending) order and then plotting the sorted values against their rescaled ranks. Irrespective of the nature of the (numerical) data, the result is always a monotonic pattern describing the function $g : R \rightarrow [0, 1]$ (of course, the domain can be a subset of R, and the function is strictly monotonic if there are not statistical ties (i.e. identical values)). If the aim is to fit a simple analytical function to such curves, theoretical cumulative distribution functions (CDFs) seem to be natural candidates. However, CDFs are not simple curves useful for fitting data, but represent the nonexceedance probability of a random variable and work if the data are independent and identically distributed (iid). All these concepts are trivial and the Authors know them better than me.

However, since daily stream flow records surely do not fulfill any of these conditions, why should a single distribution fit FDCs? In other words, in spite of the efforts made along the years to find suitable CDFs for modeling FDCs, the problem is ill-posed by definition: even the most parametrized CDFs cannot mimic FDCs unless the flow series is characterized by strong mixing (e.g. weak seasonal pattern compared to non-seasonal (essentially "random") fluctuations).

RESPONSE: We fully agree with the reviewer that daily flow series are neither independent nor identically distributed, and thus a search for a single CDF for modeling FDC's may be ill-posed, in a theoretical sense. However, there has been a continuing discussion of exactly the same issues concerning fitting of a single frequency distribution to flood series which often arise from many different (non-identical) physical mechanisms such as cyclonic, frontal and other meteorological processes. Yet it is still common practice to make the iid assumption for flood series. Similar arguments could be made for stochastic models of precipitation, drought and other hydrologic variables. Nevertheless, we agree with the reviewers concern, especially for daily streamflow which are far more complex than flood series. Thus our revised manuscript will include a section which both discusses the ill-posed nature of this important practical problem, and points to numerous approaches for deriving a composite distribution of daily streamflow based on a mixture of various processes, and we will review the literature to ensure that our work is put in the proper context. That is, this is the first comprehensive analysis of the ability of a single CDF to represent FDC's, thus it makes sense for us to summarize our results and to provide a broader context for the problem to enable others to follow up on our findings. The manuscript already provides a very brief discussion of a few studies which have sought to derive a distribution of daily streamflow from a physically based watershed model. We will use those studies to frame the 'ill-posed' nature of the problem and to provide direction for future research.

RESPONSE: In the reviewer's opinion, the ill-posed nature of our problem is both a settled and damning hypothesis; however, this is not the case within a broader scientific context. For the problem under consideration, a parsimonious, yet empirical approach to the estimation of the FDC can have profound practical implications for estimation of FDC's and even for estimation of daily streamflow series at ungaged locations. This has been shown in numerous previous studies and the textbook "Runoff Prediction in Ungauged Basins" devotes an entire chapter to predicting of FDCs at unagaged locations.

So, if the FDCs analysis reduces to a simple exercise of curve fitting, the overall analysis performed in this type of studies can make sense; otherwise, if the aim is to fit a CDF, and then concluding that such model describe probability of (non)exceedance or

something like that, this statement can be much more problematic, unless the model is a mixture of CDFs describing data approximately 'identically distributed' (id) such as seasonal or monthly subsets. In fact, the analysis reported by e.g. Basso et al. (2015) is performed on a seasonal basis.

RESPONSE: This comment reflects a possible misunderstanding of our work as being only a "curve-fitting exercise." We feel the reviewer's confusion may arise from the fact that we have not explicitly made clear the motivations for our work which involve numerous applications of FDC's which stem from the findings of this paper. Although we note this in the introduction (lines 10-11), the revised manuscript will address this issue more fully. For example, there are numerous applications of FDCs that require a complete analytical model of the FDC to implement. A common approach for estimation of daily streamflow series at ungaged sites is based on transfer of streamflow information via the CDF of streamflow from a gaged site to a nearby ungaged site. This method, has found to be a significant improvement over numerous other methods for estimation of time series of daily streamflow at ungaged sites, yet it depends on the assumption of a single pdf of streamflow. For an example of one of the first applications of this approach, using a lognormal distribution of daily streamflows, see Fennessey and Vogel (1990). We will also clearly acknowledge in the revised manuscript that our contribution is largely of a practical nature and may not satisfy one's theoretical scientific curiosity relating to the true underlying probability distribution of daily streamflow.

More generally, as the Authors know, stream flows are characterized by two properties that play a fundamental role in this context: seasonality and persistence (often long range persistence; see e.g. Montanari et al. (1997,2000) or more recently Serinaldi and Kilsby (2015)). Seasonality is often the main source departure from id condition. This is well known for instance in rainfall modeling where simple 2-parameter Weibull distributions are surely insufficient to describe daily rainfall over the entire year, but their performance is very good if we introduce parameters varying with the seasonality. Indeed the fact that stream flow values can cover two or three orders of magnitude simply

depends (obviously) from the alternation of high-flow and low-flow seasons, in which the id hypothesis is far from being realistic. On the other hand, long-range dependence results in inter-annual variability, which is what the index-flood method attempts to take into account in quite a naïve way. However, the index-flood still overlooks the problem of non-id conditions within calendar or water year. When the seasonal signal is strong, this can be the main reason for the lack of fitting of simple parametric distributions, and index-flood cannot improve the fitting very much. Moreover, while seasonality impacts on the overall shape of flow distribution (imagine to mix e.g. 12 different distributions, each reproducing approximately id monthly flows), long range dependence induces inter-annual fluctuations that impact especially on the tails. Therefore, the index-flood method adjusts more easily tail behavior than the overall shape of the parametric FDC.

RESPONSE: The reviewer raises numerous constructive points. Our revised manuscript will include an analysis which evaluates the degree to which breaking up the year into seasons can be used to improve our ability to select and model the probability distribution of daily streamflow. In addition we will add a general discussion of these issues and will point to studies which have considered these issues for analogous problems such for flood and drought problems. These comments and the seasonal analysis described, will be made in an effort to steer the reader in a direction for improvements in future studies.

RESPONSE: However, we still feel our analysis, which assumes iid conditions, (or possibly breaks up the year into seasons), is a reasonable assumption for this initial paper, given that (1) a single pdf is now used widely in regional FDC studies and (2) this is exactly the assumption which is made in practice for a very wide range of other hydrologic problems, which the reviewer mentions, including rainfall over a wide range of temporal scales, and of course floods and low flows. In other words, a solution which may not be scientifically correct is widely employed in the interest of obtaining parsimonious models which, to a first approximation, can be used to solve a very wide range of practical problems.

The above remarks, can help to understand how to improve FDC if we want to avoid physical approaches à la Botter (...but overlooking physical arguments is never a good choice) and keep the model purely statistical, but a little bit more coherent with the nature of the data. The easiest approach is surely splitting data at e.g. seasonal scale. On the other hand, we can build on the fact that the regionalization procedure commonly applied in hydrology (and summarized in this study) is only a rough and naïve version of generalized linear/additive models (GLM/GAM an their extensions) f(y; _(X)), where f is the distribution of flows Y , _ is a vector of parameters (e.g., the three parameters of the Generalized Pareto) and X is a design matrix of covariates (e.g., the variables in Eqs. 7-9). In this framework, seasonality can easily be introduced by simple sine and cosine functions describing the seasonal cycles; since a couple of waves are generally sufficient to describe the seasonal flow regime, GLMs imply only a couple of additional parameters. Alternatively, a factor index can be used in the fitting procedure to distinguish e.g. between the four seasons or the 12 months. In all cases, the resulting model not only account for the spatial variability but also for the non-id conditions by a few additional parameters that have a clear physical interpretation (they represent the seasonal regimes across the area of study).

RESPONSE: We agree that this is a useful comment and will revise our manuscript as described above. We will examine the impact of breaking the year into seasons to examine regional differences in our ability to fit a distribution to the observations. We suspect that when we break the year into four seasons, that a two-parameter Generalized Pareto distribution will fit nicely in all four seasons, resulting in four GPA distributions for a total of eight parameters. Such an analysis would be a very nice addition to our paper and would extend our conclusions and provide others with many new opportunities for research in the future involving mixtures of distributions for daily streamflow.

Of course, the usual graphical representation (as in Fig. 1) is possible only if we compare observations and simulations because such a diagrams merge quantiles coming from a set of distributions (devised for id data), roughly speaking one for each season

(or month). However, this is not surprising because the observed FDCs themselves incorporate values coming from different (seasonal) distributions, thus explaining the lack of fit of simple models.

RESPONSE: Figure 1 is simply illustrating annual flow duration curves along with their median and mean annual counterparts. Such figures are now nearly ubiquitous in hydrology, and their interpretation is quite useful in practice regardless, or in spite of, the comments of the reviewer.

This approach also helps overcoming the problem of MA-FDC simulation mentioned in the paper. Notice that the effect of seasonal variation as well as long range dependence can be recognized in Figs. 7(a-b) and 8(b-c) in the form of multimodality, while the stepwise pattern in some regions of the FDCs in Fig. 7(a) and 8(a) denotes the presence of statistical ties, which generally results from limits in the resolution of measurement devices or round-off procedures. The first aspect denotes the intrinsic inadequacy of whatever classical unimodal distribution, while the latter often affects estimation procedures (so, I'm not so surprised about the poor fitting). In this respect I have to say that the scale of the x-axis does not help fitting assessment. I'm a bit surprised because after Vogel and Fennessey (1994), we know that stretched axes enhancing the linearity of FDCs and CDFs allow much better assessment, in agreement with recommendations available in the literature on visual perception and data visualization (see e.g., works by Tufte, Cleveland, etc.).

Figures 7 and 8 both employ a logarithmic scale for streamflow on the y axis, but the reviewer is apparently suggesting the use of a 'stretched axis' for the exceedance probababilities using perhaps the inverse of a normal quantile, so that the resulting plots become equivalent to quantile-quantile lognormal plots. We elected to use the more common approach which plots the logarithms of streamflow versus exceedance probability using an ordinary arithmetic scale, because this is by far the most common approach to the graphical illustration of FDC's in practice.

Another concern is about the comparison of POR-FDCs and MA-FDC. The Authors conclude that fitting MA-FDCs is easier and more reliable than POR-FDCs as "prediction of POR-FDCs was less consistent" (consistent?). The comparison between MA-FDCs and POR-FDCs is ill-posed by itself and in the interpretation of NSE. Firstly, for MA-FDC, we always fit a CDF on 365 data points, where each one is the median (or mean) of a set of M values, where M is the number of years (here 40-60); for POR-FDCs we are trying to fit a CDF on 365 • M values (i.e. a sample 40-60 times larger), where each values (order statistics) should be the point estimates of the corresponding _ quantiles. In the first case, we seek the fitting in the range of probabilities 1 365+1 _ 3 • 10−3, 365 365+1 _ 0.997 _ , whereas in the second we pretend to fit quantiles corresponding to probabilities between 1 365M+1 _ 5 •10−5 and 365M 365M+1 _ 0.99995. So, is it so surprising that fitting a curve on 365 "smoothed" values (medians) is easier than on 18250 values (being already aware that such values cannot come, by definition, from a unique distribution)? We agree with the reviewer that comparisons between the goodness of fit of a pdf to POR-FDC and MA-FDC's is problematic due to the reasons outlined above. However, MA-FDC's are used widely for problems in which ones interest focuses on streamflow conditions in a typical or atypical year, thus it is very important for us to consider this case in our paper. The revised manuscript will include a detailed discussion of the issues raised by the reviewer and will drop all comparisons of goodness of fit between MA-FDC's and POR-FDC's and treat them as separate problems.

Secondly, the above remark allows some reflection on the (mis)use of performance metrics and their interpretation. As for every performance index (absolute metrics, relative errors, deviance or similarity measures, information criteria, etc.), NSE (which is simply the similarity index corresponding to the mean squared error) is devised to compare the performance of a set of models for the same data set; in our case, not only the sample size of the data sets and error terms is completely different (365 against about 18250), but also the nature of the data is completely incomparable (raw data against medians resulting from a very specific selection procedure). Thus, stating

that NSE for MA-FDC is generally smaller than that of POR-FDCs is nonsense, as we are comparing apples with pears. Moreover, even though I know that hydrologists have fallen in love with NSE for some esoteric reason, I would like to stress that a performance index should be chosen according to the particular type of discrepancy one wants to highlight, and not because it is popular. To be more specific, NSE is a similarity index comparing the errors from the selected model (numerator) with those from a benchmark or reference model (denominator), where the reference model is, in this case, the sample average (aka 'reference climatology' in climatological literature or "naïve" reference in forecasting literature...it seems that people in each discipline like renaming the same concepts many times, just to increment a little bit the already widespread confusion...). The choice of this "naïve" reference has two consequences: (1) the range of possible NSE values is strongly asymmetric, and (2) every model more complex than the simple average easily yields relatively high NSE values; this is usually interpreted as a good performance, but actually it is not, because the way NSE values populate the range $(-1, 1)$ is strongly nonlinear. Since the average is not a sufficient statistics even for data coming from a Gaussian distribution, it is easy to recognize that whatever model provides great improvement and (relatively high NSE) compared to such "naïve" reference. Therefore, sentences such as "Despite this comparable fit, the NSE coefficients are quite different: 0.89 for POR-FDC GPA3 versus the much higher 0.96 for MA-FDC GPA3. This discrepancy reflects a challenge in the use of the metric and indicates why visual inspection of FDC plots is particularly important for understanding overall GOF", make little sense because (1) the two values refer to different data sets (comparisons can be done only between at-site and regional models for the same data set, MA and POR, respectively), and (2) even if they referred to different models for the same data set, NSE is not equipped with criteria allowing to say if the difference between two values is significant or not (unlike methods based on maximum likelihood and/or information criteria). Concerning the rationale, choice and interpretation of performance measures please see Dawson et al. (2007), Hyndman and Koehler (2006), Jachner et al. (2007), Burnham and Anderson (2004), Reusser et

al. (2009), among others.

RESPONSE: We agree with the comments of the reviewer and, as a result, there will be no comparisons of the goodness-of-fit between the MA-FDC's and the POR-FDC's due to the reasons outlined and the revised manuscript will make this point very clearly. The use of NSE, a standardized form of mean square error, is perhaps the most commonly used goodness-of-fit metric in hydrology. We will only report log space values of NSE to deal with the fact that this goodness of fit statistic has very poor sampling properties when used with highly skewed samples as is the case for daily streamflow when NSE is computed in real space. We will continue to report these values for each case to enable comparisons of the goodness-of-fit of either MA-FDC's or POR-FDC's.

Technical remarks Please use homogeneous notation: "2-,3-,4-parameter distributions" or "two-,three- ,four-parameter distributions" throughout the text.

P3L16: it can be worth citing Doulatyari et al (2005), Basso et al. (2015), and Schaefli et al. (2013)

P6L10-15: the Authors refer to other quantile estimators; however, Weibull plotting position is not a quantile estimator. In this respect, it can also be worth having a look at Makkonen (2006), and Hutson (2000)

P7L8: "Hosking and Wallis 1997"

P7L16: "natural logarithm"

P7L16: "linear combination of order statistics" can better reflect their actual rationale (linear combination with weighted moments is a consequence)

P8L16: "see e.g. Rianna et al. (2011) and references therein"

P9L10-15: I may have missed something, but I cannot see where the effect of sample size on L-moment scattering is shown. Moreover, the similarity between L-moments

RESPONSE: Thank you for these technical comments, we will address them in the

revised paper.

References Cited:

Archfield, S.A., 2009, Chapter 2 – The Probability Distribution of Daily Streamflow, in: Estimation of continuous daily streamflow at ungaged locations in southern New England, PhD Dissertation. Tufts University.

Fennessey, N. and R.M. Vogel, Regional Flow Duration Curves for Ungaged Sites in Massachusetts, ASCE, Journal of Water Resources Planning and Management, Vol. 116, No. 4, pp. 530-549, 1990.

Vogel, R.M. and N.M. Fennessey, L-Moment Diagrams Should Replace Product-Moment Diagrams, Water Resources Research, Vol. 29, No. 6, pp 1745-1752, 1993.

---

## Short Comment (SC3) · 13 Dec 2016

Anonymous Referee 3 In their paper Blum et al. applied some well-known methodologies for finding suitable probability distributions for both period-of-record (POR) and median annual (MA) Flow Duration Curves (FDCs) in a very large area, such as the conterminous US. The authors found that, for the huge number of gauges analyzed, both the 4-parameter kappa and 3-parameter generalized Pareto distributions can reasonably simulate MA-FDC, while on the contrary even more complex distributions are unable to fit completely the very complex behavior of POR-FDCs, which explicitly accounts for extreme values. Furthermore, the authors also provide an example on possible application of their results for predicting FDC in ungauged sites, by means of the

linear regression technique. While the paper does not present in my opinion any relevant novelty from the methodological point of view, the effort of the authors to fit FDCs to such a large dataset has to be underlined.

I have few minor comments about the manuscript, that I list below. I hope my comments can help to improve further the paper. Since the research does not deal with intermittent streams, and a relevant percentage of sites (170 on 590, almost 30%) was not considered into the analysis, I would suggest to slightly modify the title of the contribution, in order to make it more fitting with the content. I suggest something like this: "The probability distribution of daily streamflow in the perennial rivers of conterminous United States". Furthermore, some words would be appreciated about future research concerning intermittent streams in conterminous US.

RESPONSE: The title will be modified as suggested and the revised manuscript will include a discussion on the need for future research on intermittent streams.

Paragraph 3.1 and Figure 3: due to the huge extension of the study area and the number of catchments analyzed, it would be interesting to verify if specific distributions fit better to specific regions or other climate/catchment features. I suggest to go at least a bit into details with this point. For example (but it's just an idea) points in Figures 3A and 3B can have different colors depending on different regions (and/or other climate/catchment distinctive features).

RESPONSE:Excellent suggestion. We are adding an analysis that assesses the goodness-of-fit of the FDCs within each of 19 major hydrologic regions of the United States to supplement the nationwide results. We also intend to evaluate how climate and catchment characteristics can help explain variations in the goodness-of-fit of the various distributions. We will make a figure as you suggest for the revised manuscript and add additional text to compliment the figure.

P 11 l 18-20: I would rather say that "the selection [...] may be as challenging as [...]". However, among the theoretical advantages associated to the index flow method, there

is the fact that complexity of Kappa and GPA distributions applied to the dimensionless daily streamflow is reduced, since the parameter alpha can be achieved as a combination of the other distribution parameters (please refer to Castellarin et al., 2007). This is a very important feature for regionalization studies. I would include this comment in the discussion. RESPONSE:Noted, we will revise the manuscript to acknowledge this point.

P 16 Eq. 7: I'm confused about using BFI as an explanatory variable, since to my knowledge it should be calculated/estimated from observed/estimated streamflow. Perhaps this variable can be replaced by some others accounting for the influence of lithplogical features on streamflows

RESPONSE:This will be removed along with the entire case study due to various concerned raised by several reviewers. The BFI_AVE was employed because it is available as a grid generated by inverse-distance-weighting of base-flow index values computed at USGS streamgages for the entire United States (Falcone et al., 2010).

Falcone, J. A., D. M. Carlisle, D. M. Wolock, and M. R. Meador (2010), GAGES: A stream gage database for evaluating natural and altered flow conditions in the conterminous United States, Ecology, 91(2), 621–621, doi:10.1890/09-0889.1.

P 18 l 6-14: I acknowledge limitations and drawbacks of using POR-FDCs, but the discussion seems to me too 'biased' towards MA-FDCs. I suggest a more detailed discussion, so that also the final sentence (l 19: "MA-FDCs [...] should not be used when severe floods and droughts are of interest") is better contextualized.

RESPONSE:We did not intend for the discussion to come across as promoting MA-FDCs and will revise the manuscript accordingly.

Finally, please consider to edit the text following the suggested corrections: P 4 l 12: "When additional goodness-of-fit (GOF) metrics..." so that you can use the acronym later (from P 5 l 17 onwards) Paragraphs 2.1 and 2.2: please correct the numbering P

6 l 20: "Figure 6 illustrates the differences..." P 7 l 16: I think that the sentence "where log represents the natural log" should be moved to line 11. P 11 l 3: maybe it could be useful for the reader if authors comment a little bit more the figure, highlighting briefly why L-moment ratios simulated from WAK are less consistent than those simulated from KAP. Captions Fig. 7 and Fig. 8: it is useful to highlight that lowest, median and highest NSE values are referred to GPA probability. Figure 9 caption: I guess one number is missing concerning the number of outliers for HUC 10

RESPONSE:Thank you very much for these comments. We will address these issues in the revised manuscript.

---

## Short Comment (SC5) · 14 Dec 2016

[supplement omitted: unrelated document]

---

## Author Response (AR1)

March 23, 2016

Dear Editors,

We have made a number of major changes to the revised paper, which are summarized below:

(1) We have had the article transferred to the new type of manuscript "Cutting-edge case studies".
(2) We have removed direct comparisons between period of record and median annual flow duration curves and removed the section on regional case studies.
(3) We have added discussion about the challenges of seasonality, as well as added additional analysis and discussion assessing the distributional fits by physiographic region in the United States.
(4) We have included lower bound adjustments and explored in greater detail the three-parameter lognormal distribution.
(5) We have added two new error duration curve figures, as suggested by a referee.

Despite the limitations of using a single distribution to describe the complexity of daily streamflows, we believe that this work still provides some useful insights to inform practical applications, such as the prediction of streamflows at ungaged sites. As the paper has been largely re-structured and re-written, we apologize for not submitting a completely marked-up version of the manuscript. The magnitude of the changes made it very difficult to maintain a complete version of this nature. Please see below for the specific changes that we have made in response to each of the referee suggestions.

Thank you.

Sincerely,
Annalise Blum, Stacey Archfield and Richard Vogel

**Actions taken in response to referee reviews**

Anonymous Referee #1

The paper fits theoretical distributions to a large dataset of empirical streamflow observations covering the conterminous US. The study finds that median annual flow
duration curves (FDC), which portray flow distribution in a typical year, can be reasonably fitted to three-parameter distributions. In contrast, period of record FDC that incorporate extreme streamflow variations over numerous years cannot be appropriately fitted, even to more complex theoretical distributions. The authors explore the implications of that finding on predictions in ungauged catchments using linear regressions in case studies. Predicting streamflow signatures (particularly FDC) in ungauged basins is both extremely useful and challenging and the findings of this study are interesting and important, particularly the insight that mAFDC might be both easier to predict and C1 more practically relevant than PoRFDC. However, there are two points that I would like to see further discussed before publication, as well as the few minor comments listed below.

First, the study is an impressive effort to fit FDCs to a very large dataset of unregulated catchments – this is definitely a key contribution of the paper. However, by covering the whole conterminous US, the dataset covers a wide variety of climates, catchment characteristics and flow regimes, and it would have been interesting to explore how the fit to specific distributions varies regionally. The shape of FDCs depicts the local flow regime, which are themselves related to climate and catchment characteristics (see e.g., Botter 2013). It would be nice to see whether there is a link between flow regimes, climate/catchment characteristics and the best fitted theoretical distribution. It would also be nice to discuss how the best-fit distributions relate to the distributions that might be expected from process-based models (Botter 2007, Botter 2009, Muller 2014, Muneepeerakul 2010, etc), given the dominant flow processes in particular catchments.

Response: Thank you for these excellent suggestions. In the revised manuscript, we will add an analysis that assesses the fit of the FDCs within each of 19 major hydrologic regions of the United States to supplement the nationwide results.  We also intend to evaluate how climate and catchment characteristics relate help explain fit to the distributions.

**Action taken: We have added a new section "Goodness of fit by physiographic region" with discussion (lines 25-36 on p 8 and continuing on p 9 lines 1-14). These physiographic regions, which differentiate between areas of the US with similar physical and climate characteristics (Fenneman and Johnson, 1946). We have also added a new figure, figure 5, to illustrate how the distributions vary regionally.**

**Ref: Fenneman, N. M., and Johnson, D, W.: Physical divisions of the United States, U.S. Geological Survey, 1:7,000,000, 1946. [online] Available from: https://water.usgs.gov/GIS/metadata/usgswrd/XML/physio.xml,**

Second, while I appreciate the effort to extend an already complex and large scale analysis to prediction in ungauged basins, I would like to see more details on how the regression models were obtained (i.e. how the regression covariates were selected for Eqn 7-9), and a discussion on whether these regression models have a physical interpretation. Specifically, I am concerned about using linear regressions to estimate distribution parameters, which arguably have a more ambiguous physical interpretation as moments. Mean flow (first moment) for instance can be argued to be a linear combination of observable characteristics like mean rainfall, as per the water balance equation. The issue in regressing GPA3 parameters is that they are not linear combinations of the moments of the distribution, so using linear regressions to estimate the parameters does not allow moments to be linearly related. In other words, in this specific case, linearly regressed parameters are not compatible with a linear water balance relation on mean flow. To address this issue, please either apply linear regressions on the moments of the distributions instead of the parameters (and discuss the physical interpretation of the linear models when appropriate), or make the case that Eqn 7-9 are not incompatible with water balance principles.

[To illustrate my point on linear regressions, let's assume the simplest linear model possible, where predictions are simply taken as the mean of the observed sample (this can happen in the specific case of the water balance model above if all catchments have an identical mean rainfall). Let's say that we have a sample of three catchments with the following GPA3 parameters and mean flow (computed from the parameters):
Basin | location param | scale param| shape param | Mean
1 | 0 | 100 | 0.3 | 143
2 | 0 | 1 |0.05 | 1
3 | 0 | 20 | 0.1 | 22
The predicted mean flow in a fourth catchment obtained from the observed mean flows (i.e. by taking the mean of the mean) is 47, whereas the mean flow computed from predicted GPA3 parameters (i.e. computed from the mean value of each parameter) is 55.]

Response: We agree that assuming that the parameters are independent from one another is a simplification that could be problematic. Based on this comment and the comments of the other reviewers, we have decided to remove the case study from the revised manuscript and add to the manuscript a deeper exploration of the FDC behavior regionally and seasonally in addition to our national analysis.
**Action taken: The regional case study has been removed.**

Minor comments:
p5 l 17: please define GOF.
Response: Thanks for catching this. We will add this explanation.
**Action taken: this is now defined on p3 line 35.**

p12 l14-19: I have seen this issue most often addressed by taking the logarithm of flow quantiles before computing the NSE. Is there are reason why you preferred the selected approach?

Response: We agree that taking the logarithm of the flow quantiles before computing the NSE is a preferred method. We had originally split the data set because there were streamgages that had zero flow values and, therefore, we cold not take the logarithms of the data. We then removed these streamgages but had kept the original reporting of the goodness of fit. In the revised manuscript, we will report the NSE values of the of logarithms of the streamflow.
**Action taken: We now use "LNSE" or NSE of the natural logarithms of the flows as explained on p5 lines 21-23. Thanks for this suggestion.**

p14 l13-15: I agree that modelling errors on FDCs are best assessed graphically and appreciate the effort of showing fits for particular basins with low, median and high NSE. Error duration curves (e.g., Muller 2016) are great way of visualizing performance fits over large samples (as opposed to individual basins), and it would be informative in my opinion to display the relevant EFDCs for the whole dataset.
selected approach?
Response:  Thank you for this useful comment. We will produce this graphic in our revisions and explore the addition of this graphic to the manuscript.
**Action taken: Error duration curves for both period of record FDC (Fig. 4) and median annual FDC (Fig 6c) are now included in the manuscript. Accompanying text is included on p 8 lines 21-30 and p 10 lines 26-29.**

p.16 l3-4: I realize that concerns of overfitting regression models are to some extent addressed in the LOO cross validation analysis, but please display covariates statistics (e.g., quartiles and range or boxplot) to show that there is enough variability in the samples to credibly argue that the LOO performance is externally valid.
Response: This is a good point, however this will be removed with the case study.
**Action taken: The regional case study has been removed.**

p.17 l2-4: Have you tested for serial correlation? Serial correlation affects the estimation of OLS standard errors and are particularly likely to occur between flow-connected gauges (i.e. gauges located on the same river).
Response: We thank the author for pointing out this question; we had not examined if there are nested basins in our dataset that may affect the OLS standard errors. We are removing the case study now from the analysis.
**Action taken: The regional case study has been removed.**

p.18 l.11-20: The paper makes the great case that MAFDCs are easier to fit and have more practical relevance than PoRFDC. However, I would appreciate a more complete discussion of the tradeoff involved: mAFDC loses information on inter-annual variability, hence their better fit to "simpler" distributions. This is an important caveat that has strong implications for practical applications and should be made clearer in the discussion/conclusion in my opinion.
Response: We agree and this change will make the revised manuscript much clearer.
**Action taken: We have removed direct comparisons between period of record and median annual flow duration curves and added text to clarify the challenges associated with median annual flow duration curves. (See p 11, Lines 13-18).**

**New text: We caution users of FDC$_{MED}$ to be aware that the FDC$_{MED}$ can only provide a window into the behavior of streamflow in a typical year, thus we recommend that whenever FDC$_{MED}$ are used that users also illustrate the entire family of annual FDCs which gave rise to the computation of the FDC$_{MED}$**

Table C1: I have trouble understanding how the BFI_AVE is a regression covariate for prediction in ungauged basins. My understanding is that flow observations are necessary to compute the BFI in the first place.

Response: The BFI_AVE is available as a grid generated by inverse-distance-weighting of base-flow index values computed at USGS streamgages for the entire United States (Falcone et al., 2010). We are removing the case study now from the analysis.

Falcone, J. A., D. M. Carlisle, D. M. Wolock, and M. R. Meador (2010), GAGES: A stream gage database for evaluating natural and altered flow conditions in the conterminous United States, *Ecology*, *91*(2), 621–621, doi:10.1890/09-0889.1.

**Action taken: The regional case study has been removed.**

Anonymous Referee 3

In their paper Blum et al. applied some well-known methodologies for finding suitable probability distributions for both period-of-record (POR) and median annual (MA) Flow Duration Curves (FDCs) in a very large area, such as the conterminous US. The authors found that, for the huge number of gauges analyzed, both the 4-parameter kappa and 3-parameter generalized Pareto distributions can reasonably simulate MA-FDC, while on the contrary even more complex distributions are unable to fit completely the very complex behavior of POR-FDCs, which explicitly accounts for extreme values. Furthermore, the authors also provide an example on possible application of their results for predicting FDC in ungauged sites, by means of the linear regression technique. While the paper does not present in my opinion any relevant novelty from the methodological point of view, the effort of the authors to fit FDCs to such a large dataset has to be underlined.

I have few minor comments about the manuscript, that I list below. I hope my comments can help to improve further the paper. Since the research does not deal with intermittent streams, and a relevant percentage of sites (170 on 590, almost 30%) was not considered into the analysis, I would suggest to slightly modify the title of the contribution, in order to make it more fitting with the content. I suggest something like this: "The probability distribution of daily streamflow in the perennial rivers of conterminous United States". Furthermore, some words would be appreciated about future research concerning intermittent streams in conterminous US.

Response: The title will be modified as suggested as well as a comment about the need for future research on intermittent streams.

**Action taken: We have changed the title to be "In search of the probability distribution of daily streamflow" to reflect the broad goals of the paper but indicate that it is an on-going search to include all types of streams. We have also added text (p 11 lines 29-32) to highlight the need for future research concerning intermittent streams.**

**New text: While there is some existing literature on intermittent regimes (Mendicino and Senatore, 2013; Pumo et al., 2014; Rianna et al., 2011), and the impacts of human regulation on flow duration curves (Gao et al., 2009; Kroll et al., 2015), additional research on these topics would improve our understanding of flows across a wider range of streams.**

Paragraph 3.1 and Figure 3: due to the huge extension of the study area and the number of catchments analyzed, it would be interesting to verify if specific distributions fit better to specific regions or other climate/catchment features. I suggest to go at least a bit into details with this point. For example (but it's just an idea) points in Figures 3A and 3B can have different colors depending on different regions (and/or other climate/catchment distinctive features).

Response: Excellent suggestion. We are adding an analysis that assesses the fit of the FDCs within each of 19 major hydrologic regions of the United States to supplement the nationwide results. We also intend to evaluate how climate and catchment characteristics relate to help explain fit to the distributions. We will make a figure as you suggest for the revised manuscript and add additional text to compliment the figure.

**Action taken: We have added a new section "Goodness of fit by physiographic region" with discussion (lines 25-36 on p 8 and continuing on p 9 lines 1-14). We have also added a new figure, figure 5, to illustrate how the distributions vary regionally.**

P 11 l 18-20: I would rather say that "the selection [...] may be as challenging as [...]". However, among the theoretical advantages associated to the index flow method, there is the fact that complexity of Kappa and GPA distributions applied to the dimensionless daily streamflow is reduced, since the parameter alpha can be achieved as a combination of the other distribution parameters (please refer to Castellarin et al., 2007). This is a very important feature for regionalization studies. I would include this comment in the discussion.

Response: Noted, we will revise the manuscript to acknowledge this point.

**Action taken: This text has been removed from the revised manuscript.**

P 16 Eq. 7: I'm confused about using BFI as an explanatory variable, since to my knowledge it should be calculated/estimated from observed/estimated streamflow. Perhaps this variable can be replaced by some others accounting for the influence of lithplogical features on streamflows

Response: This will be removed with the case study. It was used as the BFI_AVE is available as a grid generated by inverse-distance-weighting of base-flow index values computed at USGS streamgages for the entire United States (Falcone et al., 2010).

Falcone, J. A., D. M. Carlisle, D. M. Wolock, and M. R. Meador (2010), GAGES: A stream gage database for evaluating natural and altered flow conditions in the conterminous United States, *Ecology*, *91*(2), 621–621, doi:10.1890/09-0889.1.

**Action taken: The regional case study has been removed.**

P 18 l 6-14: I acknowledge limitations and drawbacks of using POR-FDCs, but the discussion seems to me too 'biased' towards MA-FDCs. I suggest a more detailed discussion, so that also the final sentence (l 19: "MA-FDCs [...] should not be used when severe floods and droughts are of interest") is better contextualized.

Response: We did not intend for the discussion to come across as promoting MA-FDCs and will revise the manuscript accordingly.

**Action taken: We have removed direct comparisons between period of record and median annual flow duration curves and added text to clarify the challenges associated with median annual flow duration curves. (See p 11, Lines 13-18)**

**New text: We caution users of $FDC_{MED}$ to be aware that the $FDC_{MED}$ can only provide a window into the behavior of streamflow in a typical year, thus we recommend that whenever $FDC_{MED}$ are used that users also illustrate the entire family of annual FDCs which gave rise to the computation of the $FDC_{MED}$.**

Finally, please consider to edit the text following the suggested corrections:

P 4 l 12: "When additional goodness-of-fit (GOF) metrics..." so that you can use the acronym later (from P 5 l 17 onwards)

Paragraphs 2.1 and 2.2: please correct the numbering

P 6 l 20: "Figure 6 illustrates the differences..."

P 7 l 16: I think that the sentence "where log represents the natural log" should be moved to line 11.

P 11 l 3: maybe it could be useful for the reader if authors comment a little bit more the figure, highlighting briefly why L-moment ratios simulated from WAK are less consistent than those simulated from KAP.

Captions Fig. 7 and Fig. 8: it is useful to highlight that lowest, median and highest NSE values are referred to GPA probability.

Figure 9 caption: I guess one number is missing concerning the number of outliers for HUC 10

Response: Thank you very much for these comments. We will address these issues in the revised manuscript.

**Action taken: Most of the text referenced here has been removed from the revised manuscript. We have added the definition of the goodness-of-fit (GOF) acronym earlier in the text (p 3, line 35).**

Reviewer #2 Dr. Serinaldi

General comments

Reviewer comments: In this paper, the Authors perform a large scale analysis in order to identify a parametric distribution function providing reasonable approximation for flow duration curves (FDCs) across the conterminous United States. The paper relies on classical "weapons" in the "statistical" arsenal commonly applied in hydrology (L-moments, Nash-Sutcliffe performance index, linear regression in logarithmic space for regionalization, etc.). So, taking for granted that such tools are sound and correctly applied, the interest in this paper is not surely methodological, but concerns the empirical results. Considering that downloading data and analyzing them with R packages such as lmom, lmomco and some build-in regression functions is a matter of few hours (most of which are needed to slightly customize the default diagrams yielded by R), in my opinion, it is a bit hard to classify this kind of works as research papers. My very personal opinion, is that they can be at most technical reports or case studies (likely resulting from some master thesis).

Response: The reviewer's comments are very far from the truth concerning the effort involved in an empirical study of this type. The 'empirical analysis' which was performed in this paper began as empirical analyses in Figures 2 and 3 in the paper by Vogel and Fennessey (1993) which only examined the pdf of daily streamflow for 23 sites in Massachusetts. To date, the paper by Vogel and Fennessey (1993) has 346 google scholar citations, and arose entirely from the empirical challenge described in the manuscript under consideration by HESS. Furthermore, the process of fitting a probability distributions to daily streamflow observations led to numerous challenges associated with zero occurrences, and obtaining plausible estimates of the lower bound of daily streamflow. It is only through such empirical studies that researchers can become aware of many of the very practical (and perhaps mundane to the reviewer) challenges and concerns in hydrology.

**Action taken: We have added text to clarify the value of this work in the context of existing literature. See new and modified text below**

**P 2 lines 28-34: Despite these theoretical and practical challenges, there is a relatively large literature which has sought to approximate the distribution of daily streamflow with a single probability distribution for very practical purposes. The main motivations have been estimation of FDCs at ungaged sites, often based on an index-flow method (Castellarin et al., 2004, 2007; Fennessey and Vogel, 1990; Li et al., 2010; Mendicino and Senatore, 2013; Rianna, 2011; Viola et al., 2011) or for estimation of time series of daily streamflow at ungaged sites (Fennessey, 1994; Smatkin and Masse, 2000; Archfield and Vogel, 2010).**

**P 11 lines 2-6 Previous work on this subject has identified the need for at least four-**

**parameters to describe the complex distribution of daily streamflows; however, this study is unique in that the suitability of a probability distribution for streamflow is investigated at the sub-continental scale with streamgages in widely-varying physiographic and hydroclimatic settings.**

**P 11 lines 17-19 Few previous studies have sought to evaluate theoretical probability distributions for modelling FDC$_{MED}$, however, their growing use suggests that our findings relating to FDC$_{MED}$ could have broad applications.**

Reviewer comments: Anyway, I leave the paper classification to the Editors; from my side, I can only say that I cannot see significant insights, while there are some inconsistencies resulting in misleading conclusions. Just to make an example, there are works providing L-moments for gridded rainfall worldwide with quite limited insight about the nature of rainfall (e.g., Maeda et al., 2013), while others (e.g. Papalexiou and Koutsoyiannis, 2016) make similar analysis on gauged data but considering distribution families derived by entropy maximization, introducing a new test for seasonal variation, and providing a number of new insights. What I mean is that we can analyze a large data set "passively", by running e.g. R codes quite blindly, or we can decide to use data in order to understand the underlying processes in more depth. Said that, if we accept the first approach, the paper is ready to publish, once removed some nonsense discussed below (concerning the comparison of MA-FDC and POR-FDC); in the second case, we are far from a good quality work. In any case, I would like to use this opportunity to share my point of view on flow duration curves (FDC), stressing that the philosophy behind them probably needs some rethinking.

Response: We welcome the detailed and insightful comments on our work, however, we note a tremendous gap between the personal thinking on this problem of this reviewer and the vast literature on FDC's. In spite of the fact that daily streamflows are certainly not independent, and probably not identically distributed, FDC's have found widespread use in practice, and numerous studies have shown that the assumption of a fixed pdf for daily streamflow can lead to improvements in our ability to estimate streamflows at ungaged sites and numerous other applications. Thus, as a practical matter, in spite of the fact that daily streamflows may not really arise from an iid process, there are many advantages of making the empirical assumption of an identically distributed process.

We encourage theoretical work by the reviewer and others to explore the possibility of modeling the probability distribution of daily streamflow as a mixture of several pdfs, however we remain quite confident that our empirical explorations of the distribution of daily streamflow may be quite useful in practice, particularly for the case of prediction in ungauged basins. It is this motivation - prediction at ungaged basins - that motivated this analysis, as the search for parsimonious solutions to fundamental problems in hydrology will always remain of use in our field. Furthermore, we note that this particular paper is our first attempt to summarize more recent investigations mentioned above which are the result of several decades of research on this topic. Thus we take great exception to numerous comments of Serinaldi when he appears to denigrate the empirical nature of work on FDC's.

We appreciate the recommendation of the author concerning Papalexiou and Koutsoyiannis (2016) which serves to further motivate the importance and relevance of our work. We will add a discussion of Papalexiou and Koutsoyiannis (2016) to our revised paper.

**Action taken: We have included additional discussion of the challenges of seasonality as well as added mention of the work of Papalexiou and Koutsoyiannis (2016) in the discussion section – see relevant text below:**

**P 11 lines 28-35: Finally, the seasonality of daily streamflows suggests that distributional analyses of this nature should be done at a seasonal level, as was recently carried out on a broad scale for daily precipitation (see Papalexiou and Koutsoyiannis, 2016). The definition of seasons, as well as the parent distributions which can approximate streamflows within those seasons, has been shown to vary across sites (Bowers et al., 2012). Given that gages**

**varied over a large range of hydroclimatic conditions, a seasonal analysis was beyond the scope of this study, but we recommend that future studies consider the impact of seasonality on the GOF of FDCs.**

Specific comments
Reviewer comments: The Authors stress twice in the text that FDC (actually POR-FDC) "ignore the important serial stochastic structure of daily flows, including such issues as autocorrelation and seasonality" and they also recognize that simple 3- or 4-parameter distributions can only approximate FDCs. These statements are given in passing, but they are actually the core of the problem. In principle we can get whatever time series of numerical values, arranging it in ascending (descending) order and then plotting the sorted values against their rescaled ranks. Irrespective of the nature of the (numerical) data, the result is always a monotonic pattern describing the function g : R ! [0, 1] (of course, the domain can be a subset of R, and the function is strictly monotonic if there are not statistical ties (i.e. identical values)). If the aim is to fit a simple analytical function to such curves, theoretical cumulative distribution functions (CDFs) seem to be natural candidates. However, CDFs are not simple curves useful for fitting data, but represent the nonexceedance probability of a random variable and work if the data are independent and identically distributed (iid). All these concepts are trivial and the Authors know them better than me.

However, since daily stream flow records surely do not fulfill any of these conditions, why should a single distribution fit FDCs? In other words, in spite of the efforts made along the years to find suitable CDFs for modeling FDCs, the problem is ill-posed by definition: even the most parametrized CDFs cannot mimic FDCs unless the flow series is characterized by strong mixing (e.g. weak seasonal pattern compared to non-seasonal (essentially "random") fluctuations).

Response: We fully agree with the reviewer that daily flow series are neither independent nor identically distributed, and thus a search for a single CDF for modeling FDC's may be ill-posed, in a theoretical sense. However, there has been a continuing discussion of exactly the same issues concerning fitting of a single frequency distribution to flood series which often arise from many different (non-identical) physical mechanisms such as cyclonic, frontal and other meteorological processes.  Yet it is still common practice to make the iid assumption for flood series. Similar arguments could be made for stochastic models of precipitation, drought and other hydrologic variables.  Nevertheless, we agree with the reviewers concern, especially for daily streamflow which are far more complex than flood series. Thus our revised manuscript will include a section which both discusses the ill-posed nature of this important practical problem, and points to numerous approaches for deriving a composite distribution of daily streamflow based on a mixture of various processes, and we will review the literature to ensure that our work is put in the proper context.  That is, this is the first comprehensive analysis of the ability of a single CDF to represent FDC's, thus it makes sense for us to summarize our results and to provide a broader context for the problem to enable others to follow up on our findings.   The manuscript already provides a very brief discussion of a few studies which have sought to derive a distribution of daily streamflow from a physically based watershed model. We will use those studies to frame the 'ill-posed' nature of the problem and to provide direction for future research.

In the reviewer's opinion, the ill-posed nature of our problem is both a settled and damning hypothesis; however, this is not the case within a broader scientific context. For the problem under consideration, a parsimonious, yet empirical approach to the estimation of the FDC can have profound practical implications for estimation of FDC's and even for estimation of daily streamflow series at ungaged locations. This has been shown in numerous previous studies and the textbook "Runoff Prediction in Ungauged Basins" devotes an entire chapter to predicting of FDCs at unagaged locations.

**Action taken: We have added discussion of the assumptions and limitations of this work. Please see relevant text below:**

**P 11 lines 19-24: There are many limitations of this work. First, daily streamflows are not independent, and thus exhibit an extremely high level of serial correlation which will impact the confidence intervals or any other form of uncertainty analysis associated with the modeled FDCs. Furthermore, daily streamflows exhibit seasonality so that they are far from being identically distributed, which is assumed whenever one attempts to fit a single**

**distribution to a random variable.**

Reviewer comments: So, if the FDCs analysis reduces to a simple exercise of curve fitting, the overall analysis performed in this type of studies can make sense; otherwise, if the aim is to fit a CDF, and then concluding that such model describe probability of (non)exceedance or something like that, this statement can be much more problematic, unless the model is a mixture of CDFs describing data approximately 'identically distributed' (id) such as seasonal or monthly subsets. In fact, the analysis reported by e.g. Basso et al. (2015) is performed on a seasonal basis.

Response:  This comment reflects a possible misunderstanding of our work as being only a "curve-fitting exercise."  We feel the reviewer's confusion may arise from the fact that we have not explicitly made clear the motivations for our work which involve numerous applications of FDC's which stem from the findings of this paper. Although we note this in the introduction (lines 10-11), the revised manuscript will address this issue more fully.  For example, there are numerous applications of FDCs that require a complete analytical model of the FDC to implement.  Common approach for estimation of daily streamflow series at ungaged sites is based on transfer of streamflow information via the CDF of streamflow from a gaged site to a nearby ungaged site. This method, has found to be a significant improvement over numerous other methods for estimation of time series of daily streamflow at ungaged sites, yet it depends on assumption of a single pdf of streamflow.  For an example of one of the first applications of this approach, using a lognormal distribution of daily streamflows, see Fennessey and Vogel (1990).   We will also clearly acknowledge in the revised manuscript that our contribution is largely of a practical nature and may not satisfy one's theoretical scientific curiosity relating to the true underlying probability distribution of daily streamflow.

**Action taken: We have added text to clarify the practical nature of our goals for this work and the numerous possible applications. Please see relevant text below:**

**P 2 lines 16-22: Historically, most studies predicting FDC$_{POR}$ at ungaged sites have used statistical methods, such as regression and index-flow methods,  due to their parsimony and relative ease of use in operational hydrology (Castellarin et al., 2013). Yet, daily streamflow observations exhibit a very high degree of serial correlation, seasonality and other complexities and are thus neither independent nor identically distributed. Klemeš (2000) warned that ignoring these complexities can be problematic, particularly if the FDC$_{POR}$ is used to extrapolate upper tails of the distribution.**

**And, in the discussion:**
**P 10 lines 31-33: Due to the complexity associated with time series of daily streamflows, the challenge set forth in this study—to identify a single probability distribution that could approximate the distribution of daily flows—was an ambitious one.**
**P 11 lines 1-2: Many assumptions were made which should be evaluated before applying these results to a practical application, as is the case for any model.**

Reviewer comments: More generally, as the Authors know, stream flows are characterized by two properties that play a fundamental role in this context: seasonality and persistence (often long range persistence; see e.g. Montanari et al. (1997,2000) or more recently Serinaldi and Kilsby (2015)). Seasonality is often the main source departure from id condition. This is well known for instance in rainfall modeling where simple 2-parameter Weibull distributions are surely insufficient to describe daily rainfall over the entire year, but their performance is very good if we introduce parameters varying with the seasonality. Indeed the fact that stream flow values can cover two or three orders of magnitude simply depends (obviously) from the alternation of high-flow and low-flow seasons, in which the id hypothesis is far from being realistic. On the other hand, long-range dependence results in inter-annual variability, which is what the index-flood method attempts to take into account in quite a naïve way. However, the index-flood still overlooks the problem of non-id conditions within calendar or water year. When the seasonal signal is strong, this can be the main reason for the lack of fitting of simple parametric distributions, and index-flood cannot improve the fitting very much. Moreover, while

seasonality impacts on the overall shape of flow distribution (imagine to mix e.g. 12 different distributions, each reproducing approximately id monthly flows), long range dependence induces inter-annual fluctuations that impact especially on the tails. Therefore, the index-flood method adjusts more easily tail behavior than the overall shape of the parametric FDC.

Response: The reviewer raises numerous constructive points. Our revised manuscript will include an analysis which evaluates the degree to which breaking up the year into seasons can be used to improve our ability to select and model the probability distribution of daily streamflow. In addition we will add a general discussion of these issues and will point to studies which have considered these issues for analogous problems such for the flood and drought problems. These comments and the seasonal analysis described, will be made in an effort to steer the reader in a direction for improvements in future studies.

However, we still feel our analysis, which assumes iid conditions, (or possibly breaks up the year into seasons), is a reasonable assumption for this initial paper, given that (1) a single pdf is now used widely in regional FDC studies and (2) this is exactly the assumption which is made in practice for a very wide range of other hydrologic problems, which the reviewer mentions, including rainfall over a wide range of temporal scales, and of course floods and low flows. In other words, a solution which may not be scientifically correct is widely employed in the interest of obtaining parsimonious models which, to a first approximation, can be used to solve a very wide range of practical problems.

**Action taken: Thank you for this excellent suggestion, however after conducting some analysis on dividing flows up by season and month, we and determined that including seasonal analysis was beyond the scope of this paper. Bowers et al (2012) explored seasonal analysis for eight rivers in the US and found large differences between them. Thus, given our goals to provide a broad analysis of nearly 400 stream gages across the US, we felt that seasonal analysis would be better left for future work that could provide more in-depth analysis of this important topic.**

**Bowers, M. C., Tung, W. W. and Gao, J. B.: On the distributions of seasonal river flows: Lognormal or power law? Water Resour. Res., 48(5), 1–12, doi:10.1029/2011WR011308, 2012.**

Reviewer comments: The above remarks, can help to understand how to improve FDC if we want to avoid physical approaches à la Botter (...but overlooking physical arguments is never a good choice) and keep the model purely statistical, but a little bit more coherent with the nature of the data. The easiest approach is surely splitting data at e.g. seasonal scale. On the other hand, we can build on the fact that the regionalization procedure commonly applied in hydrology (and summarized in this study) is only a rough and naïve version of generalized linear/additive models (GLM/GAM an their extensions) f(y; _(X)), where f is the distribution of flows Y , _ is a vector of parameters (e.g., the three parameters of the Generalized Pareto) and X is a design matrix of covariates (e.g., the variables in Eqs. 7-9). In this framework, seasonality can easily be introduced by simple sine and cosine functions describing the seasonal cycles; since a couple of waves are generally sufficient to describe the seasonal flow regime, GLMs imply only a couple of additional parameters. Alternatively, a factor index can be used in the fitting procedure to distinguish e.g. between the four seasons or the 12 months. In all cases, the resulting model not only account for the spatial variability but also for the non-id conditions by a few additional parameters that have a clear physical interpretation (they represent the seasonal regimes across the area of study).

Response: We agree that this is a useful comment and will revise our manuscript as described above. We will examine the impact of breaking the year into seasons to examine regional differences in the ability to fit a distribution to the observations. We suspect that when we break the year into four seasons, that two-parameter Generalized Pareto distribution will fit nicely in all four seasons, resulting in four GPA distributions for a total of eight parameters. Such an analysis would be a very nice addition to our paper and would extend our conclusions and provide others with many new opportunities for research in the future involving mixtures of distributions for daily streamflow.

**Action taken: This is an important point, however, for the reasons noted above, we deemed this beyond the scope of the current study.**

Of course, the usual graphical representation (as in Fig. 1) is possible only if we compare observations and simulations because such a diagrams merge quantiles coming from a set of distributions (devised for id data), roughly speaking one for each season (or month). However, this is not surprising because the observed FDCs themselves incorporate values coming from different (seasonal) distributions, thus explaining the lack of fit of simple models.

Response:  Figure 1 is simply illustrating annual flow duration curves along with their median and mean annual counterparts. Such figures are now nearly ubiquitous in hydrology, and their interpretation is quite useful in practice regardless, or in spite of, the comments of the reviewer.

**Action taken: This figure has been removed from the revised manuscript.**

Reviewer comments: This approach also helps overcoming the problem of MA-FDC simulation mentioned in the paper. Notice that the effect of seasonal variation as well as long range dependence can be recognized in Figs. 7(a-b) and 8(b-c) in the form of multimodality, while the stepwise pattern in some regions of the FDCs in Fig. 7(a) and 8(a) denotes the presence of statistical ties, which generally results from limits in the resolution of measurement devices or round-off procedures. The first aspect denotes the intrinsic inadequacy of whatever classical unimodal distribution, while the latter often affects estimation procedures (so, I'm not so surprised about the poor fitting). In this respect I have to say that the scale of the x-axis does not help fitting assessment. I'm a bit surprised because after Vogel and Fennessey (1994), we know that stretched axes enhancing the linearity of FDCs and CDFs allow much better assessment, in agreement with recommendations available in the literature on visual perception and data visualization (see e.g., works by Tufte, Cleveland, etc.).

Figures 7 and 8 both employ a logarithmic scale for streamflow on the y axis, but the reviewer is apparently suggesting the use of a 'stretched axis' for the exceedance probababilities using perhaps the inverse of a normal quantile, so that the resulting plots become equivalent to quantile-quantile lognormal plots. We elected to use the more common approach which plots the logarithms of streamflow versus exceedance probability using an ordinary arithimetic scale, because this is by far the most common approach to the graphical illustration of FDC's in practice.

Another concern is about the comparison of POR-FDCs and MA-FDC. The Authors conclude that fitting MA-FDCs is easier and more reliable than POR-FDCs as "prediction of POR-FDCs was less consistent" (consistent?). The comparison between MA-FDCs and POR-FDCs is ill-posed by itself and in the interpretation of NSE. Firstly, for MA-FDC, we always fit a CDF on 365 data points, where each one is the median (or mean) of a set of M values, where M is the number of years (here 40-60); for POR-FDCs we are trying to fit a CDF on $365 \cdot M$ values (i.e. a sample 40-60 times larger), where each values (order statistics) should be the point estimates of the corresponding _ quantiles. In the first case, we seek the fitting in the range of probabilities $\frac{1}{365+1}$ _ $3 \cdot 10-3$, $\frac{365}{365+1}$ _ $0.997$
_ , whereas in the second we pretend to fit quantiles corresponding to probabilities between $\frac{1}{365M+1}$ _ $5 \cdot 10-5$ and $\frac{365M}{365M+1}$ _ $0.99995$. So, is it so surprising that fitting a curve on 365 "smoothed" values (medians) is easier than on 18250 values (being already aware that such values cannot come, by definition, from a unique distribution)?

Response: We agree with the reviewer that comparisons between the goodness of fit of a pdf to POR-FDC and MA-FDC's is problematic due to the reasons outlined above.  However, MA-FDC's are used widely for problems in which ones interest focuses on streamflow conditions in a typical or atypical year, thus it is very important for us to consider this case in our paper. The revised manuscript will include a detailed discussion of the issues raised by the reviewer and will drop all comparisons of goodness of fit between MA-FDC's and POR-FDC's and treat them as separate problems.

**Action taken: We have removed direct comparisons between period of record and median annual flow duration curves and all discussion of median annual flow duration curve results is now in section 4.4. We have also added the text:**
**P 11, Lines 13-18: We caution users of FDC$_{MED}$ to be aware that the FDC$_{MED}$ can only provide a window into the behavior of streamflow in a typical year, thus we recommend that whenever FDC$_{MED}$ are used that users also illustrate the entire family of annual FDCs which gave rise to the computation of the FDC$_{MED}$.**

Secondly, the above remark allows some reflection on the (mis)use of performance metrics and their interpretation. As for every performance index (absolute metrics, relative errors, deviance or similarity measures, information criteria, etc.), NSE (which is simply the similarity index corresponding to the mean squared error) is devised to compare the performance of a set of models for the same data set; in our case, not only the sample size of the data sets and error terms is completely different (365 against about 18250), but also the nature of the data is completely incomparable (raw data against medians resulting from a very specific selection procedure). Thus, stating that NSE for MA-FDC is generally smaller than that of POR-FDCs is nonsense, as we are comparing apples with pears. Moreover, even though I know that hydrologists have fallen in love with NSE for some esoteric reason, I would like to stress that a performance index should be chosen according to the particular type of discrepancy one wants to highlight, and not because it is popular. To be more specific, NSE is a similarity index comparing the errors from the selected model (numerator) with those from a benchmark or reference model (denominator), where the reference model is, in this case, the sample average (aka 'reference climatology' in climatological literature or "naïve" reference in forecasting literature...it seems that people in each discipline like renaming the same concepts many times, just to increment a little bit the already widespread confusion...). The choice of this "naïve" reference has two consequences: (1) the range of possible NSE values is strongly asymmetric, and (2) every model more complex than the simple average easily yields relatively high NSE values; this is usually interpreted as a good performance, but actually it is not, because the way NSE values populate the range (−1, 1) is strongly nonlinear. Since the average is not a sufficient statistics even for data coming from a Gaussian distribution, it is easy to recognize that whatever model provides great improvement and (relatively high NSE) compared to such "naïve" reference. Therefore, sentences such as "Despite this comparable fit, the NSE coefficients are quite different: 0.89 for POR-FDC GPA3 versus the much higher 0.96 for MA-FDC GPA3. This discrepancy reflects a challenge in the use of the metric and indicates why visual inspection of FDC plots is particularly important for understanding overall GOF", make little sense because (1) the two values refer to different data sets (comparisons can be done only between at-site and regional models for the same data set, MA and POR, respectively), and (2) even if they referred to different models for the same data set, NSE is not equipped with criteria allowing to say if the difference between two values is significant or not (unlike methods based on maximum likelihood and/or information criteria). Concerning the rationale, choice and interpretation of performance measures please see Dawson et al. (2007), Hyndman and Koehler (2006), Jachner et al. (2007), Burnham and Anderson (2004), Reusser et al. (2009), among others.

Response: We agree with the comments of the reviewer and, as a result, there will be no comparisons of the goodness-of-fit between the MA-FDC's and the POR-FDC's due to the reasons outlined and the revised manuscript will make this point very clearly. The use of NSE, a standardized form of mean square error, is perhaps the most commonly used goodness-of-fit metric in hydrology. We will only report log space values of NSE to deal with the fact that this goodness of fit statistic has very poor sampling properties when used with highly skewed samples as is the case for daily streamflow in real space.  We will continue to report these values for each case to enable comparisons of the goodness-of-fit of either MA-FDC's or POR-FDC's.

**Action taken: We have removed direct comparisons between period of record and median annual flow duration curves and have replaced real-space NSE estimates with those computed from natural logarithms of the flows as explained on p5 lines 21-23. Thanks for this suggestion.**

Technical remarks
Please use homogeneous notation: "2-,3-,4-parameter distributions" or "two-,three- ,four-parameter

distributions" throughout the text. **Action taken: Change was made.**

P3L16: it can be worth citing Doulatyari et al (2005), Basso et al. (2015), and Schaefli et al. (2013) **Action taken: We have added citations to these papers.**

P6L10-15: the Authors refer to other quantile estimators; however, Weibull plotting position is not a quantile estimator. In this respect, it can also be worth having a look at Makkonen (2006), and Hutson (2000) **Action taken: We have clarified the text to address this point.**

P7L8: "Hosking and Wallis 1997" **Action taken: Change was made.**

P7L16: "natural logarithm" **Action taken: Change was made.**

P7L16: "linear combination of order statistics" can better reflect their actual rationale (linear combination with weighted moments is a consequence) **Action taken: Change was made.**

P8L16: "see e.g. Rianna et al. (2011) and references therein" **Action taken: The text has change to include additional references cited in Rianna et al. (2011).**

P9L10-15: I may have missed something, but I cannot see where the effect of sample size on L-moment scattering is shown. Moreover, the similarity between L-moments **Action taken: We have modified the text in an attempt to clarify this section.**

Thank you for these technical comments, we will address them in the revised paper.

References Cited:

Archfield, S.A., 2009, Chapter 2 – The Probability Distribution of Daily Streamflow, in: Estimation of continuous daily streamflow at ungaged locations in southern New England, PhD Dissertation. Tufts University.

Fennessey, N. and R.M. Vogel, Regional Flow Duration Curves for Ungaged Sites in Massachusetts, ASCE, Journal of Water Resources Planning and Management, Vol. 116, No. 4, pp. 530-549, 1990.

Vogel, R.M. and N.M. Fennessey, L-Moment Diagrams Should Replace Product-Moment Diagrams, Water Resources Research, Vol. 29, No. 6, pp 1745-1752, 1993.

---

## Author Response (AR2)

May 26, 2016

Dear Editors,

We have made the technical corrections requested by the Referee. Please find the comments, responses, and marked-up manuscript below.

With regards to the title, we have changed it to: "On the probability distribution of daily streamflow in the United States". We believe that the addition of "On the" indicates that we are contributing the larger challenge of finding one distribution for the whole United States. We feel that adding the word conterminous to the title would detract from the clarity of the title and one will quickly learn that the study is of the conterminous United States upon
reading the abstract.

Also, we wanted to highlight that the author order has changed since our original submission (to Blum, Archfield and Vogel.)

Thank you for all your work with us on this manuscript.

Sincerely,

Annalise Blum, Stacey Archfield, and Richard Vogel

**Referree comments and actions taken**

**Comments: Suggestions for revision or reasons for rejection (will be published if the paper is accepted for final publication)** Authors have deeply modified the original version of the paper, taking into account many of the reviewers' comments and concerns.

Furthermore, the paper is proposed as a case study, which is fair given that no substantial methodological innovations are proposed. In its current version, the manuscript represents a basis for further detailed analyses on single regions, providing useful information about the most suitable distributions for both PORFDC and MEDFDC across conterminous US. Future researches listed at the end of the manuscript (P. 12, LL. 5-7) should contribute to improve significantly the current findings highlighted by the authors.

Please consider the following minor corrections/typos:

Title: Probably it's better "In search of the probability distribution of daily streamflow in the conterminous US" **Action taken: We have added "On the" at the beginning and "in the**

**United States" at the end of the title.**

P. 2, L. 33: Rianna et al., 2011 **Action taken: Thank you for catching this, it has been corrected.**

P. 2, L. 38: to describe **Action taken: Thank you for catching this, it has been corrected.**

P. 10, L. 15: "Then, the median flow at each ranking is selected for inclusion FDCMED."

Not clear **Action taken: This sentence has been revised and now reads: "**Then, the median flow at each ranking is selected to represent the given quantile within the $FDC_{MED}$**"**

P. 10, L. 17: probabilities **Action taken: Thank you for catching this, it has been corrected.**

P. 11, L. 13: I guess "Cascade Sierra regions" **Action taken: Thank you for catching this, it has been corrected.**

P. 11, L. 21: "…we recommend that whenever FDCMED are used that users…" please check **Action taken: This sentence has been revised and now reads: "**
[revised manuscript text omitted]